# Etiologies and 12-month mortality in patients with isolated involuntary weight loss at a rapid diagnostic unit

Jordi Aligué[1,2]*, Mireia Vicente[3], Anna Arnau[4], Jaume Trapé[2,5], Eva Martínez[6], Mariona Bonet[6], Andrés Abril[3], Omar El Boutrouki[6], Roser Ordeig[7], Domingo Ruiz[2,6], Josep Ordeig[6], Antonio San José[8]

1 Central Catalonia Chronicity Research Group (C3RG), Internal Medicine Department, Althaia Xarxa Assistencial Universitària de Manresa, Manresa, Spain, 2 Faculty of Medicine, University of Vic—Central University of Catalonia (UVIC-UCC), Vic, Spain, 3 Primary Care, Institut Català de la Salut, Barcelona, Spain, 4 Central Catalonia Chronicity Research Group (C3RG), Clinical Research Unit. Althaia Xarxa Assistencial Universitària de Manresa, Manresa, Spain, 5 Department of Laboratory Medicine, Althaia Xarxa Assistencial Universitària de Manresa, Manresa, Spain, 6 Internal Medicine Department, Althaia Xarxa Assistencial Universitària de Manresa, Manresa, Spain, 7 Primary Care, Althaia Xarxa Assistencial Universitària de Manresa, Manresa, Spain, 8 Internal Medicine Service, Hospital Universitari Vall d'Hebron, Universitat Autònoma de Barcelona, Barcelona, Spain

* jaligue@althaia.cat

**Data Availability Statement:** All relevant data are within the paper and its Supporting Information files.

**Funding:** The authors received no specific funding for this work.

## Abstract

### Background

Numerous studies on involuntary weight loss (IWL) have been published since the 1980s, although most of them have included small samples of patients with specific symptoms. The aim of the present study was to determine the causes, demographic and clinical characteristics and mortality at 12 months in patients attended at a rapid diagnostic unit (RDU) for isolated IWL.

### Methods

A single-center retrospective observational study including all patients presenting to the RDU for isolated IWL between 2005 and 2013. The following data were recorded: demographic and clinical variables, results of complementary tests (blood tests, x-rays, computed tomography scan and digestive endoscopy), main diagnosis and vital status at 12 months.

### Results

Seven hundred and ninety-one patients met the criteria for IWL. Mean age was 67.9 years (SD 4.7), 50.4% were male and mean weight loss was 8.3 kg (SD 4.7). The cause for IWL was malignant disease in 23.6% of patients, non-malignant organic disease in 44.5%, psychiatric disorder in 29.0% and unknown in 3.2%. Overall mortality at 12 months was 18.6% (95%CI: 16.1–21.6). The mortality rate was highest in the group with malignancy (61.1%; 95%CI: 54.2–68.2).

**Competing interests:** The authors have declared that no competing interests exist.

## Conclusions

Almost a quarter of all patients attended at the RDU for IWL were diagnosed with cancer. Mortality at 12 months was higher in this group than in the other three. Malignancy should therefore be ruled out during the first visit for patients attended for IWL.

## Introduction

Involuntary weight loss (IWL) is defined as a reduction of ≥5% of the usual body weight in a period of six months or less [1] although most authors agree on prolonging this period to 12 months [2, 3]. In frail elderly people, smaller losses may also be important [4]. IWL is frequently a sign of underlying illness [5, 6] and is associated with increased morbidity and mortality [1].

Most studies published on IWL include small samples of patients with specific symptoms that may help to guide further investigations in particular areas. Few studies have focused on isolated weight loss in larger samples [3–5, 7–10]. What is more, studies on patients with weight loss are habitually performed in hospitalized patients; today, however, outpatient rapid diagnostic units (RDUs) allow us to optimize resources and achieve high levels of satisfaction among both patients and providers [11, 12]. At these RDUs, the most frequently diagnosed pathology is cancer (18–30%) [13, 14]. Few studies have been carried out entirely in patients with isolated IWL treated in RDUs [15], who may present different etiologies from those described in studies performed in outpatients visits or in hospitalized patients.

Studies published in the past 20 years [3, 7, 15–19] report the following ranges of prevalence of causes of weight loss: organic or non-malignant 32 to 51%, cancer 6 to 35%, psychiatric 11 to 34%, and unknown etiology 6 to 28%.

The aim of the present study was to determine the causes, demographic and clinical characteristics, and 12-month mortality in patients attended at a RDU for isolated IWL.

## Materials and methods

We designed a retrospective observational study of patients attended at an RDU between January 2005 and December 2013 for isolated IWL. Follow-up was completed in December 2014. Our RDU is an outpatient resource that allows the study of severe pathologies without the need for hospital admission. It caters for patients with signs and symptoms warranting referral from other hospital services or primary care. Criteria for referral to the RDU are shown in the S1 File. RDUs have highly specific diagnostic algorithms based on signs and symptoms warranting referral. In the case of isolated IWL, however, the recommendations are very general: anamnesis, physical examination, a complete blood test and chest X-ray. Exclusion criteria for RDU are severe functional dependency (Barthel Index≤20), cognitive decline (Global Deterioration Scale>3), lack of family support (unavailability of a main caregiver) for attending the outpatient center, and chronic symptoms (more than 12 months of evolution).

The study protocol was approved on 6 March 2015 by an independent clinical research ethics committee (Comitè d'Ètica d'Investigació de la Fundació Unió Catalana d'Hospitals—Ethical Committee number CEI 15/16). The study was exempt from the requirement for informed consent. Medical patients' records of Hospital Sant Joan de Déu de Manresa were accessed between June 2016 and June 2017.

All patients presenting with documented isolated IWL of at least 5% over the past 12 months were eligible for inclusion. IWL was considered isolated when it was not accompanied

by symptoms or signs specific to a particular organ or system. When weight loss was not documented, the criteria of Marton et al. [20] were used; i.e., patients were eligible if they met at least two of the following criteria: evidence of change in clothes size, confirmation of weight loss by a friend or relative and ability to give a numerical estimate of weight loss.

Patients with the following criteria assessed at the first RDU visit were excluded: (i) specific symptoms (jaundice, ascites, dysphagia for at least six months, diarrhea for at least six months, rectal bleeding, intestinal transit disorders, rectal tenesmus and/or suspicious rectal examination, intestinal subocclusion crises, cough that had changed for over a month, hemoptysis of unknown origin, dysphonia for at least one month, palpable breast mass, nipple discharge, macroscopic hematuria, significantly enlarged lymph nodes (>1 cm), suspicion of malignant hepatomegaly according to physical examination or diagnostic imaging prior to the initial RDU visit, metrorrhagia and abdominal mass), (ii) intentional weight loss, (iii) initiation of diuretic treatment within three months before start of IWL, (iv) weight loss of <5% or no weight loss observed during first visit at RDU, (v) refusal to participate in follow-up assessments and/or undergo further complementary tests; (vi) non-compliance with referral criteria for RDU. Patients who died during the diagnostic process and in whom it was not possible to perform the required diagnostic tests were also excluded.

The RDU database was used to identify patients. Demographic data, clinical and follow-up history were obtained from patients' clinical records. The following variables were recorded: age, gender, living situation, referral from primary care or from hospital, dates of first and last RDU visits, number of kilograms (kg) lost and time elapsed since start of IWL, smoking habit, alcohol consumption (g/day), Charlson comorbidity index, previous psychiatric history, associated unspecific symptoms (asthenia, anorexia, fever, depressive symptoms, abdominal pain, nausea and/or vomiting, arthralgias and benign lymph nodes) which were not considered as symptoms warranting referral, hemoglobin (Hb) (g/dL), leukocytes (x10e9/L), erythrocyte sedimentation rate (ESR) (mm/h), glycosylated hemoglobin (%), creatinine (mg/dL), estimated glomerular filtrate rate (eGFR) (mL/min), albumin (g/dL), aspartate aminotransferase (AST) (U/L), alanine aminotransferase (ALT) (U/L), total bilirubin (mg/dL), gamma-glutamyl transferase (GGT) (U/L), alkaline phosphatase (ALP) (U/L), iron (µg/dL), ferritin (ng/mL), thyroid-stimulating hormone (TSH) (µUI/mL), lactate dehydrogenase (LDH) (U/L), C-reactive protein (CRP) (mg/L), fecal occult blood test (FOBT), results of imaging tests (X-rays, abdominal ultrasound and thoracic and/or abdominal computed tomography (CT) scan) and digestive endoscopy (gastroscopy and/or colonoscopy).

The primary outcome was the etiology of IWL, classified as neoplasia, non-malignant organic disease, psychiatric pathology and isolated IWL of unknown origin (i.e., when not determined after 12 months of follow-up). The final diagnosis was made by consensus between two physicians at the internal medicine service. The secondary outcome measure was 12-month mortality.

## Statistical analysis

**Sample size.** The sample size required to estimate a prevalence of malignancy of 56% [21] in patients attended for IWL as symptom warranting referral, with a confidence level of 95% and a precision of ±4%, was 592 patients. Assuming that 5% of patients would be lost to follow-up, 624 patients had to be included. The required sample size was calculated using Ene 2.0 software 0 (www.e-biometria.com).

Categorical variables are presented as absolute values and relative frequencies. Continuous variables are summarized as means and standard deviations (SD) for normal distributions, and as medians and interquartile range [IQR] for non-normal distributions. For the bivariate

analysis, the one-way ANOVA was used for continuous variables. The chi-square test, Fisher's exact test or the Monte Carlo method was used for categorical variables.

One-year mortality Kaplan-Meier curves were constructed and the log-rank test was used to compare them. The level of statistical significance was 5% (two-sided p<0.05). The IBM SPSS Statistics v.22 (IBM Corporation®, Armonk, New York) and STATA v.14 (StataCorp LP®, College Station, Texas) programs were used for statistical analysis.

## Results

From January 2005 to December 2013, a total of 1592 patients were attended and assessed for IWL. Of these, 791 (49.7%) met the criteria for isolated IWL (Fig 1). The most frequent exclusion criteria were IWL with specific symptoms (68.0%) and weight reduction of <5% (24.2%).

Table 1 shows patients' baseline characteristics. Mean age was 67.9 years (SD 15.7), and 50.4% were men. Statistically significant differences (p<0.001) were observed in the distribution of sexes between the four diagnostic groups: the highest percentage of men was found in the neoplastic group (67.0%) and the highest presence of women in the psychiatric disorder group (65.1%). Most patients were referred from primary care (69.2%). In the group of patients with IWL of unknown origin, there was a higher percentage referred from primary healthcare centers. In the group of neoplastic patients, the percentage referred from the radiology department, and who had undergone endoscopic procedures, was greater (p = 0.001). Mean weight loss in the past 12 months was 8.3 Kg (SD 4.7). In the neoplastic and non-malignant organic disease groups, almost 50% had had IWL for less than three months, while in the psychiatric disorders and unknown origin groups, most had had IWL for more than six months.

The most frequent non-specific symptoms associated with IWL were anorexia (47.4%), abdominal pain (34.4%) and depressive symptoms (31.7%), while the most frequent non-specific symptom associated with malignancy was abdominal pain. Regarding habits, 24.4% were smokers and 23.5% consumed alcohol (mean 25.3 g/day, SD 46.8). Twelve percent of patients had a high comorbidity burden (Charlson Index ≥3). Patients with psychiatric disorders and IWL of unknown etiology had lower comorbidity than cancer patients and those with non-malignant organic disease. Previous psychiatric disorder (44.5%) and use of psychoactive drugs (38.9%) were significantly higher in the psychiatric disorder group (p<0.001).

Table 2 shows the analytical parameters obtained in the different groups. Patients diagnosed with cancer had significantly lower levels of hemoglobin, albumin and iron, and significantly higher levels of leukocytes, ESR, GGT and ALP, ferritin, TSH, LDH and CRP.

Imaging findings are summarized in Table 3. Of the 634 chest X-rays performed, 5.8% were suspicious for malignancy; in the cancer group, 19.2% (p<0.001) of chest X-rays presented malignancy. Of the 72 X-rays performed at other sites, 6.9% were suspicious for malignancy. This percentage was statistically higher in the neoplastic group (26.3%; p = 0.039).

Abdominal ultrasound was performed in 35.5% of patients, and 14.6% were suspicious of malignancy. In the neoplastic group, 53.1% of abdominal ultrasound were suspicious for malignancy, compared with 4.7% in the non-malignant organic disorder group, 1.2% in the psychiatric disorder group and none in the IWL of unknown origin group (p<0.001).

Suspicion of malignancy was recorded in 32.1% of thoracic CTs, 41.9% of abdominal CTs and 54.2% of thoracoabdominal CTs. In the neoplastic group, 80% of thoracic, 86.3% of abdominal and 90.7% of thoracoabdominal CTs presented suspicion of malignancy. In the non-malignant organic disorder group, 15.4% of abdominal and 11.9% of thoracoabdominal CTs showed images with suspicion of malignancy (which proved to be false positives at the end of the diagnostic process).

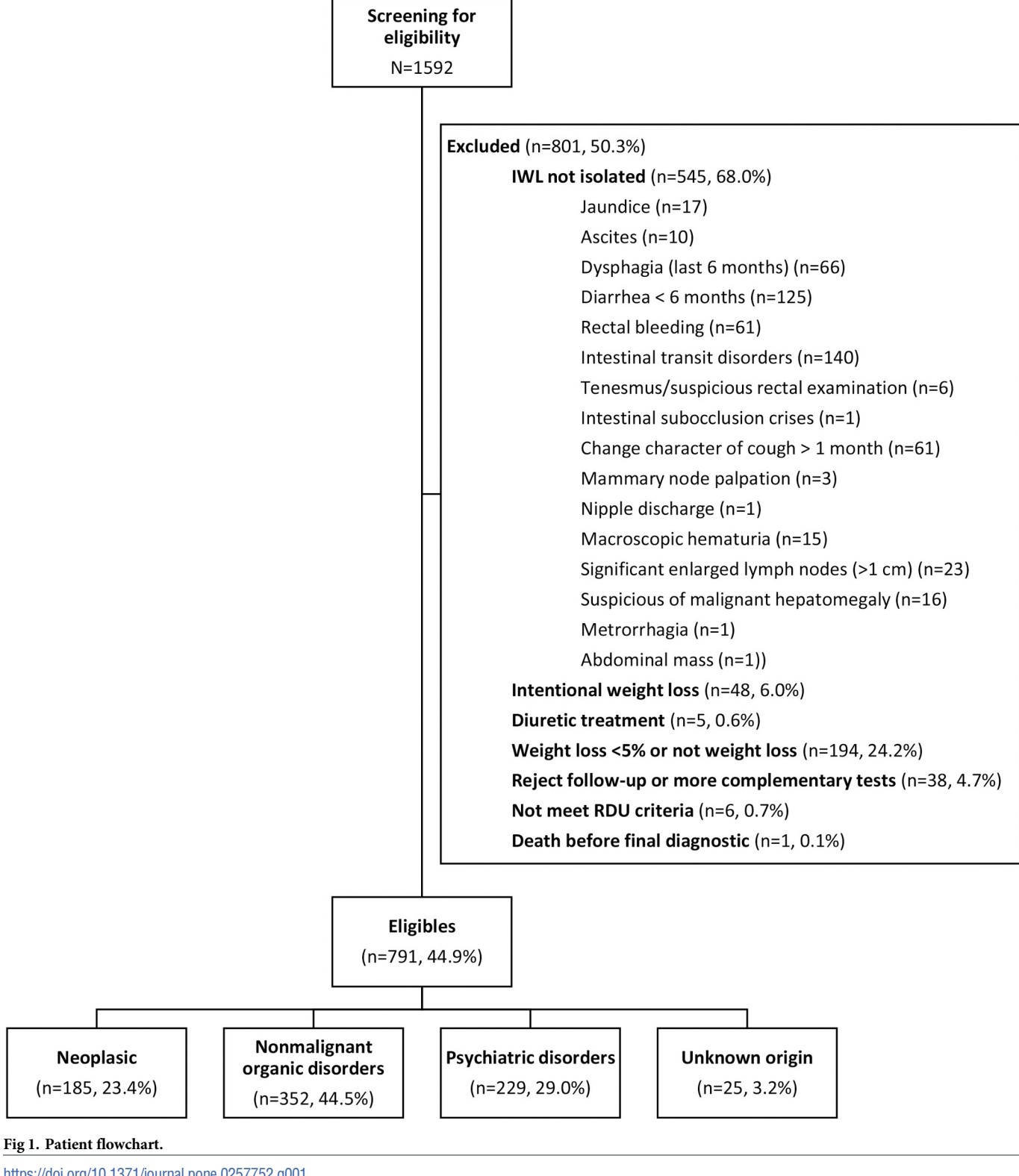

**Fig 1. Patient flowchart.**

Gastroscopy was performed in 272 patients (34.4%) and colonoscopy in 122 (15.4%).
Malignancy was histologically confirmed in 39 gastroscopies (14.3%) and in 22 colonoscopies

**Table 1. Baseline characteristics of patients with IWL, overall and according to etiology.**

| | n valid | Overall | Neoplastic | Non-malignant organic disorders | Psychiatric disorders | Unknown origin | p-value |
|---|---|---|---|---|---|---|---|
| | | **n = 791** | **n = 185** | **n = 352** | **n = 229** | **n = 25** | |
| **Age** | 791 | 67.9 (SD 15.7) | 71.5 (SD 12.1) | 68.0 (SD 15.7) | 65.6 (SD 16.9) | 60.7 (SD 22.8) | 0.017 |
| **Sex** | 791 | | | | | | <0.001 |
| Males | | 399 (50.4%) | 124 (67%) | 181 (51.4%) | 80 (34.9%) | 14 (56%) | |
| **Place of residence** | 791 | | | | | | 0.113 |
| Private residence | | 770 (97.3%) | 178 (96.2%) | 336 (95.5%) | 226 (98.7%) | 25 (100%) | |
| Residential home | | 21 (2.7%) | 7 (3.8%) | 16 (4.5%) | 3 (1.3%) | 0 (0%) | |
| **Sources of referral to RDU** | 791 | | | | | | 0.001 |
| Primary healthcare centers | | 547 (69.2%) | 124 (67%) | 242 (68.8%) | 160 (69.9%) | 21 (84%) | |
| Emergency department | | 148 (18.7%) | 40 (21.6%) | 61 (17.3%) | 43 (18.8%) | 4 (16%) | |
| Outpatient consultations | | 80 (10.1%) | 8 (4.3%) | 46 (13.1%) | 26 (11.4%) | 0 (0%) | |
| Radiology department | | 11 (1.4%) | 9 (4.9%) | 2 (0.6%) | 0 (0%) | 0 (0%) | |
| Endoscopic procedures | | 5 (0.6%) | 4 (2.2%) | 1 (0.3%) | 0 (0%) | 0 (0%) | |
| **IWL FEATURES** | | | | | | | |
| **Weight loss (kg) previous 12 months** | 775 | 8.3 (SD 4.7) | 7.9 (SD 4.3) | 8.5 (SD 5.0) | 8.4 (SD 4.6) | 7.9 (SD 3.8) | 0.793 |
| **Onset of IWL (months)** | 791 | | | | | | 0.018 |
| <1 month | | 60 (7.6%) | 10 (5.4%) | 37 (10.5%) | 12 (5.2%) | 1 (4%) | |
| between 1 and 2 months | | 127 (16.1%) | 41 (22.2%) | 54 (15.3%) | 28 (12.2%) | 4 (16%) | |
| between 2 and < 3 months | | 145 (18.3%) | 40 (21.6%) | 67 (19%) | 35 (15.3%) | 3 (12%) | |
| between 3 and <4 months | | 106 (13.4%) | 22 (11.9%) | 52 (14.8%) | 30 (13.1%) | 2 (8%) | |
| between 4 and <5 months | | 59 (7.5%) | 15 (8.1%) | 26 (7.4%) | 17 (7.4%) | 1 (4%) | |
| between 5 and <6 months | | 58 (7.3%) | 12 (6.5%) | 27 (7.7%) | 16 (7%) | 3 (12%) | |
| >6 months | | 236 (29.8%) | 45 (24.3%) | 89 (25.3%) | 91 (39.7%) | 11 (44%) | |
| **ASSOCIATED SYMPTOMS** | | | | | | | |
| **Asthenia (Yes)** | 791 | 232 (29.3%) | 62 (33.5%) | 100 (28.4%) | 63 (27.5%) | 7 (28%) | 0.553 |
| **Anorexia (Yes)** | 791 | 375 (47.4%) | 94 (50.8%) | 169 (48%) | 101 (44.1%) | 11 (44%) | 0.566 |
| **Fever (Yes)** | 791 | 48 (6.1%) | 12 (6.5%) | 31 (8.8%) | 5 (2.2%) | 0 (0%) | 0.008 |
| **Depression (Yes)** | 791 | 251 (31.7%) | 27 (14.6%) | 65 (18.5%) | 155 (67.7%) | 4 (16%) | <0.001 |
| **Abdominal pain (Yes)** | 791 | 272 (34.4%) | 80 (43.2%) | 116 (33%) | 68 (29.7%) | 8 (32%) | 0.029 |
| **Nausea and/or vomiting (Yes)** | 791 | 125 (15.8%) | 27 (14.6%) | 68 (19.3%) | 29 (12.7%) | 1 (4%) | 0.049 |
| **Arthralgia (Yes)** | 791 | 83 (10.5%) | 26 (14.1%) | 34 (9.7%) | 21 (9.2%) | 2 (8%) | 0.346 |
| **Non-pathological adenopathy (Yes)** | 791 | 1 (0.1%) | 0 (0%) | 1 (0.3%) | 0 (0%) | 0 (0%) | - |
| **HABITS** | | | | | | | |
| **Smoker** | 791 | | | | | | 0.010 |
| No | | 434 (54.9%) | 87 (47%) | 193 (54.8%) | 141 (61.6%) | 13 (52%) | |
| Yes | | 193 (24.4%) | 51 (27.6%) | 76 (21.6%) | 59 (25.8%) | 7 (28%) | |
| Former smoker | | 164 (20.7%) | 47 (25.4%) | 83 (23.6%) | 29 (12.7%) | 5 (20%) | |
| **Alcohol consumption (Yes)** | 791 | 186 (23.5%) | 66 (35.7%) | 82 (23.3%) | 31 (13.5%) | 7 (28%) | <0.001 |
| **Daily alcohol consumption (g/day)** | 186 | 25.3 (SD 46.8) | 19.5 (SD 20.4) | 26.3 (SD 47.6) | 35.8 (SD 78.5) | 22.6 (SD 24.8) | 0.454 |
| **COMORBIDITY** | | | | | | | |
| **Oral cavity problems (Yes)** | 791 | 31 (3.9%) | 9 (4.9%) | 12 (3.4%) | 9 (3.9%) | 1 (4%) | 0.902 |
| **Previous psychiatric problems (Yes)** | 791 | 197 (24.9%) | 21 (11.4%) | 70 (19.9%) | 102 (44.5%) | 4 (16%) | <0.001 |
| **Psychotropic drugs (Yes)** | 791 | 190 (24.0%) | 30 (16.2%) | 67 (19%) | 89 (38.9%) | 4 (16%) | <0.002 |
| **Charlson Comorbidity Index** | 791 | 0.99 (SD 1.3) | 1.1 (SD 1.3) | 1.1 (SD 1.3) | 0.8 (SD 1.1) | 0.6 (SD 1.2) | 0.012 |
| No comorbidity (0–1 points) | | 583 (73.7%) | 133 (71.9%) | 251 (71.3%) | 178 (77.7%) | 21 (84%) | 0.060 |

*(Continued)*

**Table 1.** (Continued)

| | n valid | Overall | Neoplastic | Non-malignant organic disorders | Psychiatric disorders | Unknown origin | p-value |
|---|---|---|---|---|---|---|---|
| | | n = 791 | n = 185 | n = 352 | n = 229 | n = 25 | |
| Low comorbidity (2 points) | | 113 (14.3%) | 25 (13.5%) | 53 (15.1%) | 35 (15.3%) | 0 (0%) | |
| High comorbidity (≥3 points) | | 95 (12.0%) | 27 (14.6%) | 48 (13.6%) | 16 (7%) | 4 (16%) | |
| **Hospital admittance in last 12 months (Yes)** | 760 | 139 (18.3%) | 35 (19.6%) | 64 (19.1%) | 37 (16.7%) | 3 (12.5%) | 0.730 |

n valid: number of patients with valid data

(18%). In the neoplastic group, 50% of gastroscopies and 53.7% of colonoscopies were histologically confirmed as malignant.

Table 4 presents the data referring to the etiology of IWL. The cause of isolated IWL was: cancer in 23.4%, non-malignant organic disease in 44.5%, psychiatric disorder in 29.0% and unknown cause in 3.2%. The most common cancer was digestive (12.0%), followed by lung (3.5%). In the patients with digestive cancer, the most frequent forms were gastric (4.8%),

**Table 2. Blood and stool analysis parameters of patients with IWL, overall and according to etiology.**

| | n valid | Overall | Neoplastic | Non-malignant organic disorders | Psychiatric disorders | Unknown origin | p-value |
|---|---|---|---|---|---|---|---|
| | | n = 791 | n = 185 | n = 352 | n = 229 | n = 25 | |
| **Hb** (g/dL) | 788 | 13.1 [11.9–14.3] | 12.2 [10.8–13.9] | 13.2 [12.1–14.3] | 13.4 [12.4–14.4] | 12.7 [11.7–14.7] | <0.001 |
| **Leukocytes** (x10e9/L) | 784 | 7.5 [6.2–9.4] | 8.3 [6.9–10.1] | 7.4 [6.0–9.3] | 7.2 [5.8–8.8] | 6.9 [6.0–9.0] | <0.001 |
| **ESR** (mm/h) | 699 | 28 [14–55] | 52 [30–81] | 28 [14–54] | 19 [9–31] | 20 [9–44] | <0.001 |
| **Glucose** (mg/dL) | 775 | 100 [92–114] | 104 [96–123] | 100 [92–116] | 98 [90–107] | 99 [88–116] | <0.001 |
| **Glycated hemoglobin** (%) | 130 | 6.2 [5.6–7.0] | 6.5 [5.6–7.5] | 6.3 [5.6–7.0] | 6.0 [5.6–6.4] | 5.7 [4.6–6.0] | 0.142 |
| **Creatinine** (mg/dL) | 778 | 0.9 [0.7–1.0] | 0.9 [0.7–1.1] | 0.9 [0.7–1.1] | 0.8 [0.7–1] | 0.8 [0.7–1] | 0.010 |
| **eGFR** (mL/min) | 780 | | | | | | 0.805 |
| Stage 1 and 2: > 60 | | 606 (77.7%) | 140 (75.7%) | 266 (75.6%) | 178 (77.7%) | 22 (88%) | |
| Stage 3: 30–60 | | 158 (20.3%) | 41 (22.2%) | 70 (19.9%) | 44 (19.2%) | 3 (12%) | |
| Stage 4: 15–29 | | 13 (1.7%) | 2 (1.1%) | 8 (2.3%) | 3 (1.3%) | 0 (0%) | |
| Stage 5: <15 | | 3 (0.4%) | 1 (0.5%) | 2 (0.6%) | 0 (0%) | 0 (0%) | |
| **Albumin** (g/dL) | 696 | 3.9 [3.4–4.1] | 3.5 [3.1–3.9] | 3.9 [3.5–4.1] | 4 [3.7–4.3] | 4 [3.7–4.3] | <0.001 |
| **AST** (U/L) | 754 | 21 [17–27] | 22 [17–33] | 22 [18–29] | 20 [17–24] | 20 [19–24] | <0.001 |
| **ALT** (U/L) | 742 | 18 [13–26] | 18 [13–27] | 18 [14–29] | 17 [13–22] | 16.5 [14–25] | 0.065 |
| **Total bilirubin** (mg/dL) | 751 | 0.8 [0.6–1.0] | 0.7 [0.6–1.0] | 0.8 [0.6–1.1] | 0.8 [0.5–1.0] | 0.6 [0.5–0.9] | 0.041 |
| **GGT** (U/L) | 756 | 21 [14–42] | 29 [17–81] | 23 [15–42] | 17 [13–27] | 18 [12–29] | <0.001 |
| **ALP** (U/L) | 724 | 67 [53–88] | 80 [63–129] | 66 [52–85] | 62 [50–77] | 57 [53–73] | <0.001 |
| **Iron** (µg/dL) | 512 | 65 [40–92] | 43 [21–75] | 66 [40–95] | 74 [59–98] | 60 [52–104] | <0.001 |
| **Ferritin** (ng/mL) | 534 | 85 [33–199] | 131 [44–368] | 90 [36–223] | 59 [23–143] | 70 [31–157] | <0.001 |
| **TSH** (µUI/mL) | 620 | 1.5 [0.9–2.1] | 1.7 [1.1–2.8] | 1.4 [0.9–2.1] | 1.4 [0.9–2.0] | 1.1 [0.7–1.7] | 0.009 |
| **LDH** (U/L) | 684 | 398 [344–486] | 449 [346–693] | 400 [344–474] | 389 [341–456] | 374 [340–437] | <0.001 |
| **CRP** (mg/L) | 623 | 4,9 [1,6–24,0] | 29,2 [4,9–79,0] | 5,0 [2,0–19,8] | 2,0 [1,0–5,0] | 2,6 [1,0–10,0] | <0,001 |
| **FOBT** (Yes) | 791 | 110 (13.9%) | 16 (8.6%) | 42 (11.9%) | 47 (20.5%) | 5 (20%) | 0.003 |
| **Positive FOBT** | 110 | 35 (31.8%) | 8 (50.0%) | 14 (33.3%) | 11 (23.4%) | 2 (40.0%) | 0.240 |

Hb: hemoglobin; ESR: erythrocyte sedimentation rate; eGFR: estimated glomerular filtration rate; AST: aspartate transaminase; ALT; alanine transaminase; GGT: gamma glutamyltransferase; ALP: alkaline phosphatase; TSH: thyrotropin; LDH: lactate dehydrogenase CRP; C-reactive protein; FOBT: fecal occult blood test. Median [percentile 25-percentile 75].

**Table 3. Diagnostic imaging of patients with IWL, overall and according to etiology.**

|  | Overall | Neoplastic | Non-malignant organic disorders | Psychiatric disorders | Unknown origin | p-value |
|---|---|---|---|---|---|---|
|  | n = 791 | n = 185 | n = 352 | n = 229 | n = 25 |  |
| **X-rays (Yes)** | 654 (82.7%) | 157 (84.9%) | 279 (79.3%) | 199 (86.9%) | 19 (76%) | 1.000 |
| **Chest X-ray (Yes)** | 634 (80.2%) | 151 (81.6%) | 274 (77.8%) | 191 (83.4%) | 18 (72%) | 0.651 |
| Normal | 419 (66.1%) | 93 (61.6%) | 181 (66.1%) | 134 (70.2%) | 11 (61.1%) | <0.001 |
| Suspected malignancy | 37 (5.8%) | 29 (19.2%) | 8 (2.9%) | 0 (0%) | 0 (0%) |  |
| Other non-malignant findings | 178 (28.1%) | 29 (19.2%) | 85 (31%) | 57 (29.8%) | 7 (38.9%) |  |
| **Abdomen X-ray (Yes)** | 168 (21.2%) | 42 (22.7%) | 72 (20.5%) | 48 (21%) | 6 (24%) | 0.106 |
| Normal | 140 (83.3%) | 33 (78.6%) | 63 (87.5%) | 39 (81.3%) | 5 (83.3%) | 0.583 |
| Suspected malignancy | 1 (0.6%) | 1 (2.4%) | 0 (0%) | 0 (0%) | 0 (0%) |  |
| Other non-malignant findings | 27 (16.1%) | 8 (19%) | 9 (12.5%) | 9 (18.8%) | 1 (16.7%) |  |
| **Other X-rays (Yes)** | 72 (9.1%) | 19 (10.3%) | 31 (8.8%) | 21 (9.2%) | 1 (4%) | 0.630 |
| Normal | 30 (41.7%) | 6 (31.6%) | 13 (41.9%) | 11 (52.4%) | 0 (0%) | 0.039 |
| Suspected malignancy | 5 (6.9%) | 5 (26.3%) | 0 (0%) | 0 (0%) | 0 (0%) |  |
| Other non-malignant findings | 37 (51.4%) | 8 (42.1%) | 18 (58.1%) | 10 (47.6%) | 1 (100%) |  |
| **Abdomen ultrasound (Yes)** | 281 (35.5%) | 64 (34.6%) | 128 (36.4%) | 82 (35.8%) | 7 (28%) | 0.272 |
| Normal | 130 (46.3%) | 16 (25%) | 56 (43.8%) | 54 (65.9%) | 4 (57.1%) | <0.001 |
| Suspected malignancy | 41 (14.6%) | 34 (53.1%) | 6 (4.7%) | 1 (1.2%) | 0 (0%) |  |
| Other non-malignant findings | 110 (39.1%) | 14 (21.9%) | 66 (51.6%) | 27 (32.9%) | 3 (42.9%) |  |
| **Thoracic CT scan (Yes)** | 53 (6.7%) | 20 (10.8%) | 27 (7.7%) | 6 (2.6%) | 0 (0%) | 1.000 |
| Normal | 12 (22.6%) | 2 (10%) | 7 (25.9%) | 3 (50%) | 0 (0%) | <0.001 |
| Suspected malignancy | 17 (32.1%) | 16 (80%) | 1 (3.7%) | 0 (0%) | 0 (0%) |  |
| Other non-malignant findings | 24 (45.3%) | 2 (10%) | 19 (70.4%) | 3 (50%) | 0 (0%) |  |
| **Abdomen CT scan (Yes)** | 129 (16.3%) | 51 (27.6%) | 52 (14.8%) | 23 (10%) | 3 (12%) | 0.025 |
| Normal | 33 (25.6%) | 2 (3.9%) | 18 (34.6%) | 11 (47.8%) | 2 (66.7%) | <0.001 |
| Suspected malignancy | 54 (41.9%) | 44 (86.3%) | 8 (15.4%) | 2 (8.7%) | 0 (0%) |  |
| Other non-malignant findings | 42 (32.6%) | 5 (9.8%) | 26 (50%) | 10 (43.5%) | 1 (33.3%) |  |
| **Thoracoabdominal CT Scan (Yes)** | 155 (19.6%) | 86 (46.5%) | 42 (11.9%) | 21 (9.2%) | 6 (24%) | 0.009 |
| Normal | 23 (14.8%) | 3 (3.5%) | 12 (28.6%) | 6 (28.6%) | 2 (33.3%) | <0.001 |
| Suspected malignancy | 84 (54.2%) | 78 (90.7%) | 5 (11.9%) | 1 (4.8%) | 0 (0%) |  |
| Other non-malignant findings | 48 (31.0%) | 5 (5.8%) | 25 (59.5%) | 14 (66.7%) | 4 (66.7%) |  |
| **Gastroscopy (Yes)** | 272 (34.4%) | 78 (42.2%) | 123 (34.9%) | 67 (29.3%) | 4 (16%) | <0.001 |
| Normal | 99 (36.4%) | 20 (25.6%) | 40 (32.5%) | 38 (56.7%) | 1 (25%) | <0.001 |
| Malignancy | 39 (14.3%) | 39 (50%) | 0 (0%) | 0 (0%) | 0 (0%) |  |
| Other non-malignant findings | 134 (49.3%) | 19 (24.4%) | 83 (67.5%) | 29 (43.3%) | 3 (75%) |  |
| **Colonoscopy (Yes)** | 122 (15.4%) | 41 (22.2%) | 47 (13.4%) | 32 (14%) | 2 (8%) | 0.019 |
| Normal | 49 (40.2%) | 10 (24.4%) | 24 (51.1%) | 15 (46.9%) | 0 (0%) | <0.001 |
| Malignancy | 22 (18%) | 22 (53.7%) | 0 (0%) | 0 (0%) | 0 (0%) |  |
| Other non-malignant findings | 51 (41.8%) | 9 (22%) | 23 (48.9%) | 17 (53.1%) | 2 (100%) |  |

colonic (3.2%) and pancreatic (2.4%). The most frequent non-malignant organic cause of IWL (n = 352) was of digestive origin (16.4%), with a high percentage of peptic disease (5.2%). The most predominant pharmacological causes were digoxin (1.3%) and metformin (1.1%).

Four percent of IWL had an infectious cause, most frequently of pulmonary origin; 3.7% of IWL had a rheumatic cause, among which rheumatic polymyalgia was the most frequently diagnosed (2.4%). The most frequent psychiatric cause was depression (25.0%), followed by anxiety (2.3%). In 3.2% of patients no cause was identified after 12-month follow-up, and so they were labeled as isolated IWL of unknown origin.

**Table 4. Causes of IWL.**

| | n = 791 | |
|---|---|---|
| **Neoplastic** | **185** | **23.4%** |
| Digestive tract | 95 | 12.0% |
| Gastric | 38 | 4.8% |
| Colon | 25 | 3.2% |
| Pancreas | 19 | 2.4% |
| Liver and intrahepatic bile ducts | 7 | 0.9% |
| Retroperitoneum and peritoneum | 2 | 0.3% |
| Gall bladder and extrahepatic bile duct | 1 | 0.1% |
| Esophagus | 1 | 0.1% |
| Small intestine | 1 | 0.1% |
| Rectum, sigmoid colon and anus | 1 | 0.1% |
| Trachea, bronchus and lungs | 28 | 3.5% |
| Leukemia and myeloproliferative disorders | 14 | 1.8% |
| Multiple myeloma | 8 | 1.0% |
| Myelodysplasia syndrome | 3 | 0.4% |
| Acute myeloid leukemia and precursors to neoplasia | 2 | 0.3% |
| Myeloproliferative neoplasia | 1 | 0.1% |
| Lymphoma | 12 | 1.5% |
| Kidney and urinary tract | 10 | 1.3% |
| Female genital tract | 4 | 0.5% |
| Prostate | 3 | 0.4% |
| Breast | 3 | 0.4% |
| Neuroendocrine tumor | 1 | 0.1% |
| Thyroid glands | 1 | 0.1% |
| Thymus, heart and mediastinum | 1 | 0.1% |
| Unknown primary site | 13 | 1.6% |
| **Non-malignant organic disorders** | **352** | **44.5%** |
| Digestive system | 130 | 16.4% |
| Peptic disorders | 41 | 5.2% |
| Hiatus hernia | 18 | 2.3% |
| Chronic alcohol induced liver disease | 17 | 2.1% |
| Gallstones | 10 | 1.3% |
| Functional dyspepsia | 9 | 1.1% |
| Diverticula disorders | 6 | 0.8% |
| Barrett's esophagus | 4 | 0.5% |
| Incompetent cardia | 3 | 0.4% |
| Chronic alcohol-induced pancreatitis | 3 | 0.4% |
| Chronic hepatitis C | 2 | 0.3% |
| Large benign colon polyps treated surgically | 2 | 0.3% |
| Other disorders | 15 | 1.9% |
| **Pharmacological causes** | **45** | **5.7%** |
| Digoxin | 10 | 1.3% |
| Metformin | 9 | 1.1% |
| Opioids | 6 | 0.8% |
| Nonsteroidal anti-inflammatories | 4 | 0.5% |
| Levothyroxine | 4 | 0.5% |
| ACEI/ARB | 2 | 0.3% |

(*Continued*)

**Table 4.** (Continued)

|  | n = 791 | |
| --- | ---: | ---: |
| Others | 10 | 1.3% |
| Infections | 32 | 4.0% |
| Rheumatism | 29 | 3.7% |
| Neurological disorders | 29 | 3.7% |
| Endocrine diseases | 18 | 2.3% |
| Lung disorders | 15 | 1.9% |
| Musculoskeletal disorders | 15 | 1.9% |
| Heart disorders | 12 | 1.5% |
| Psychosocial circumstances | 10 | 1.3% |
| Excessive alcohol consumption ($\geq$30g/day in women and $\geq$40g/day in men) | 4 | 0.5% |
| Genitourinary problems | 4 | 0.5% |
| Blood disorders | 2 | 0.3% |
| Others | 7 | 0.9% |
| **Psychiatric disorders** | **229** | **29.0%** |
| Depression | 198 | 25.0% |
| Anxiety | 18 | 2.3% |
| Eating disorders | 4 | 0.5% |
| Others | 9 | 1.1% |
| **Unknown origin** | **25** | **3.2%** |

ACEI/ARB: angiotensin-converting enzyme inhibitors and angiotensin II receptor blockers.

Fig 2 shows an overall mortality at 12 months of 18.6% (95%CI: 16.1–21.6). The highest rate was recorded in the neoplastic group (61.1%; 95%CI: 54.2–68.2) followed by the non-malignant organic group (6.4%; 95%CI: 4.3–9.6), psychiatric disorder (3.6%; 95%CI: 1.8–7.1) and unknown origin (3.9%; 95%CI: 0.5–25.2) (Fig 3).

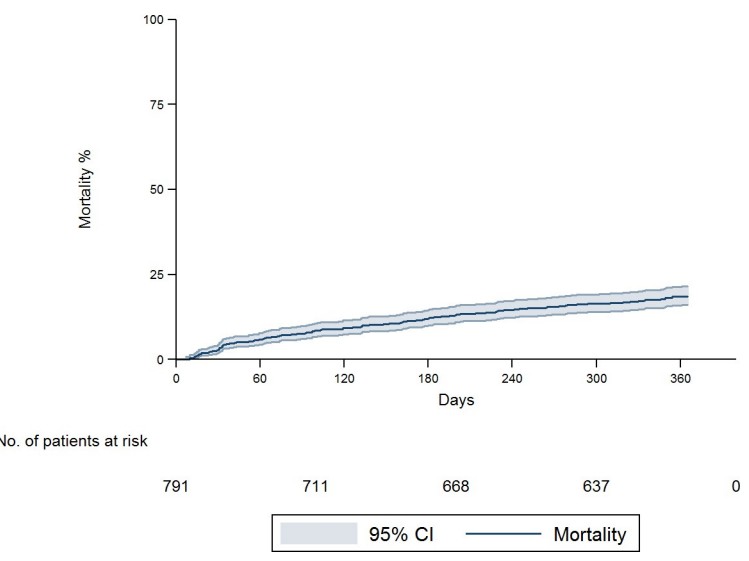

**Fig 2. Overall 12-month mortality.**

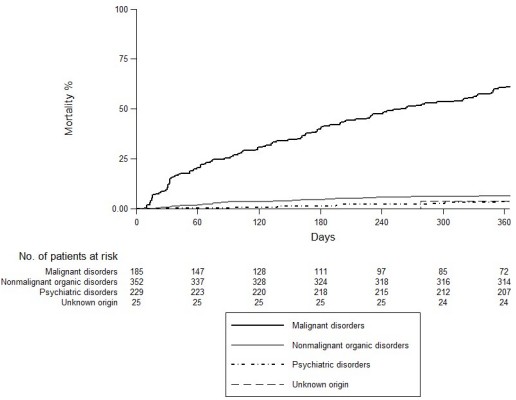

**Fig 3. Overall 12-month mortality according to etiology.**

## Discussion

Almost a quarter of all patients attended at the RDU for IWL were diagnosed with cancer. The most frequent cause of isolated IWL was non-malignant organic disease, predominantly digestive disease. Mortality at 12 months was higher in the neoplastic group than in the other three.

In our study, the prevalence of cancer in patients with involuntary weight loss was similar to that recorded in some previous studies (22–24.8%) [3, 16, 19, 22], but lower than those found in others (33–36.3%) [15, 17, 23]. These differences may be due to the inclusion of hospitalized patients in the other studies [17, 23] (evaluated outside the RDU) or, in the case of studies of IWL performed in RDUs, the inclusion of patients with other specific symptoms [15]. Consistent with other published series, malignancies were more frequent in men [10, 15, 24] and in older patients [3, 10, 15, 24].

The most frequent diagnosis in the present study was digestive cancer, which accounted for 51% of the total. Among these cancers, 40% were gastric, 26% colonic and 20% pancreatic. These findings are in line with those of previous studies performed with similar samples (45–54%) [3, 10, 15].

The most frequent cause of IWL in our sample was non-malignant organic disease (44.5%). Previous studies have found similar rates for non-malignant organic disease, ranging from 33.8% to 50.7% [3, 7, 15, 18–20, 24–26].

Psychiatric disorders (especially depression) were more frequent in women, and were found in 29% of patients with IWL, in agreement with previous studies (24–33%) [16, 17, 23]. Bilbao-Garay [16] found that the most frequent cause of IWL was psychiatric disorder, and reported a rate of depression of 67% (compared with our figure of 86.5%). In our study the prevalence of depression and anxiety in the psychiatric disorders group was around 94%, similar to the rate recorded by Bilbao-Garay et al [16].

Unlike other studies [3, 7, 15–18, 26], we found a lower percentage of IWL of unknown cause (3.2%). This may be due to the greater availability of complementary tests (both imaging tests and endoscopy), the long follow-up to confirm diagnosis, and the appraisal of the patient's psychosocial situation (an aspect not included in most studies).

In agreement with Rabinovitz et al. [23] and Metalidis et al. [3], we did not find significant differences in degree of weight loss with respect to etiology. In contrast, Bosch et al. [15] and Vierboom et al. [6] reported that cancer patients had more pronounced weight loss than other patients. However, these studies included patients not only with isolated IWL but also IWL as a dominant feature of disease.

A careful history may be very useful for localizing signs or symptoms that may guide further investigations in particular areas. However, there is no consensus on the tests that should be included in the initial evaluation of isolated IWL. Nonetheless, most authors seem to agree that a detailed medical history, a thorough physical examination, a complete blood test and chest X-ray seem to be sufficient for an initial evaluation.

The presence of anemia, low serum albumin, iron deficiency and higher levels of leukocytes, ESR, GGT, ALP, ferritin, TSH, LDH and CRP was associated with an increased risk of cancer, as in the study by Baicus et al. [18]. Therefore, any alteration of these basic analytical parameters in patients presenting with isolated IWL raises the suspicion of inflammatory processes, localized neoplastic processes, or disseminated disease. For example, ferritin may be increased in different types of cancer, GGT in cases of liver metastasis and ALP in cases of liver and/or bone metastases.

When assessing the profitability of imaging studies to detect malignancy, the following CTs had highest sensitivity: thoracoabdominal (90.7%), abdominal (86.3%) and thoracic (80%), while abdominal ultrasound showed a lower sensitivity (53.1%). Comparison with previous studies such as Hernández et al. [10] is difficult, since those authors performed CTs in only 25% of patients diagnosed with cancer, compared with the figure of 85% in our neoplastic group.

IWL was associated with an increase in mortality [1]. The overall mortality rate at 12 months was 18.2%, similar to that described by Bosch et al. [15] with a mean follow-up of 14.5 months. As expected, mortality was higher (61.1%) when the etiology of IWL was malignancy. In contrast, mortality rates at 12 months were 6.3% for non-malignant organic pathology, 3.5% for psychiatric disorder and 4% for IWL of unknown cause. These findings are also concordant with those described by Bosch et al. [15], who reported mortality rates of 69%, 6%, 5% and 5% respectively.

Most previous studies have been carried out on IWL in individuals with specific symptoms, mainly with small series of hospitalized patients. One of the strengths of the present study of isolated IWL is that it is based on the largest sample of patients (n = 791) of all articles on IWL published to date, and the second study performed in patients at a RDU.

Our study has some limitations. First, no studies are available of isolated IWL performed exclusively in primary care, where non-malignant organic disease and psychiatric disorder are likely to be more prevalent causes than in our sample. Our study was carried out at a RDU; therefore, many patients with non-malignant organic disease and psychiatric disorder may have previously been diagnosed and treated in primary care and would not have been referred to the RDU. Although Primary Care physicians may request complementary tests such as imaging tests or endoscopies, in our setting, when there is a suspicion of malignancy, patients are referred to the RDU. Consequently, pathologies such as diabetes mellitus, hyperthyroidism or depression, which can be diagnosed in Primary Care based on a clinical assessment and a standard analysis, may be more frequent in the general population with IWL than in our sample. In addition, patients with dementia or severe dependency or lack of family support could be diagnosed with a malignant disorder outside the RDU. Second, the inclusion and definition criteria used to refer patients to a RDU, and the procedures for conducting workup, may differ substantially between providers. This means that epidemiological and demographic characteristics of evaluable patients may differ between hospitals. Another source of variability may be differences in the ability of the physicians to evaluate IWL. As a result, the implications of our study cannot be directly generalized to other settings; however, other studies have found similar results in the prevalence of cancer among patients with IWL attended at RDUs. Third, 4.7% of patients were excluded without obtaining a diagnosis, either because the patient refused follow-up at the RDU or because complementary tests were not performed.

In conclusion, our results show that malignancy should be ruled out in the initial assessment of isolated IWL. Twenty-three per cent of the patients evaluated at the RDU for isolated IWL were diagnosed with cancer. Over 50% of the malignancies diagnosed were digestive (mainly gastric, colonic or pancreatic). Furthermore, mortality in the malignancy group was higher than in other patients.

Computed tomography was the most yield complementary test to be performed in addition to complete anamnesis, a complete blood test and chest X-ray for a good diagnostic approximation.

## Supporting information

**S1 File. Criteria for referral to the RDU in patients over the age of 18.**
(DOCX)

## Acknowledgments

The authors thank all the patients who took part in the study. We are particularly grateful to Sylva Torossian and Michael Maudsley for their help with the translation and editing of the manuscript.

## Author Contributions

**Conceptualization:** Jordi Aligué, Jaume Trapé, Antonio San José.

**Data curation:** Anna Arnau, Jaume Trapé, Eva Martínez, Mariona Bonet, Andrés Abril, Omar El Boutrouki, Roser Ordeig, Josep Ordeig, Antonio San José.

**Formal analysis:** Jordi Aligué, Anna Arnau, Jaume Trapé, Antonio San José.

**Investigation:** Jordi Aligué, Mireia Vicente, Anna Arnau, Jaume Trapé, Eva Martínez, Mariona Bonet, Andrés Abril, Antonio San José.

**Methodology:** Jordi Aligué, Anna Arnau, Jaume Trapé, Antonio San José.

**Project administration:** Jordi Aligué.

**Resources:** Jordi Aligué, Jaume Trapé.

**Supervision:** Mireia Vicente, Anna Arnau, Jaume Trapé, Domingo Ruiz, Josep Ordeig, Antonio San José.

**Validation:** Jordi Aligué, Mireia Vicente, Anna Arnau, Jaume Trapé, Domingo Ruiz, Antonio San José.

**Visualization:** Jordi Aligué, Mireia Vicente, Domingo Ruiz, Antonio San José.

**Writing – original draft:** Jordi Aligué, Mireia Vicente, Jaume Trapé.

**Writing – review & editing:** Jordi Aligué, Mireia Vicente, Anna Arnau, Jaume Trapé, Omar El Boutrouki, Roser Ordeig, Domingo Ruiz, Josep Ordeig, Antonio San José.

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
