## [Decision Letter · Decision Letter 0]

1 Feb 2021

PONE-D-20-05824

Etiologies and mortality at 12 months of patients with isolated involuntary weight loss in a rapid diagnostic unit

PLOS ONE

Dear Dr. Aligué,

Thank you for submitting your manuscript to PLOS ONE. After careful consideration, we feel that it has merit but does not fully meet PLOS ONE’s publication criteria as it currently stands. Therefore, we invite you to submit a revised version of the manuscript that addresses the points raised during the review process.

The manuscript has been evaluated by two reviewers, and their comments are available below. You will see the reviewers have commented on the interest of your study. However, the reviewers have also raised critical concerns and the manuscript will need significant revision before it can be considered for publication – you should anticipate that the reviewers will be re-invited to assess the revised manuscript, so please ensure that your revision is thorough. I have outlined some of the key concerns noted by the reviewers below, but you should respond all concerns mentioned by the reviewers in your response-to-reviewers document. 

The key concerns noted by the reviewers relate to requests for more information about the rationale for the study design, addition details regarding the diagnoses, clarification about the exclusion criteria, further descriptions of the study population. Additionally, please describe study limitations in the Discussion section. These issues have limitations for the interpretation of the results and should be explored.

Please note that novelty is not a requirement for publication in PLOS ONE: https://journals.plos.org/plosone/s/editorial-and-peer-review-process

We look forward to receiving your revised manuscript.

Kind regards,

Danielle Poole

Staff Editor

PLOS ONE

Journal Requirements:

2. In the ethics statement in the manuscript and in the online submission form, please provide additional information about the patient records used in your retrospective study, including: a) whether all data were fully anonymized before you accessed them; b) the date range (month and year) during which patients' medical records were accessed; and c) the source of the medical records analyzed in this work (e.g. hospital, institution or medical center name). If patients provided informed written consent to have data from their medical records used in research, please include this information.

3. At this time, we ask that you please discuss any limitations of your study in the Discussion section.

Reviewers' comments:

Reviewer's Responses to Questions

**Comments to the Author**

1. Is the manuscript technically sound, and do the data support the conclusions?

Reviewer #1: Yes

Reviewer #2: Partly

2. Has the statistical analysis been performed appropriately and rigorously? 

Reviewer #1: Yes

Reviewer #2: Yes

3. Have the authors made all data underlying the findings in their manuscript fully available?

Reviewer #1: Yes

Reviewer #2: Yes

4. Is the manuscript presented in an intelligible fashion and written in standard English?

Reviewer #1: No

Reviewer #2: Yes

5. Review Comments to the Author

Reviewer #1: Thank you for the opportunity to review this interesting study.

1. I suggest having a native English speaker review content for grammar, style, and spell check.

Examples:

Line 32: change “visited at” to “presenting to”

Line 34: spell out “computed tomography” for the firs t time “CT” is used.

Line 40: “… which was higher in the malignant group” is awkward phrasing – suggest “…with the mortality rate being the highest in those with malignancy (61.1%, 95% CI 54.2,68.2)

Line 57: pluralize “rapid diagnostic unit”, or, if singular, then on line 56, add “an” before “ambulatory”

Line 58: “users”: this should be possessive (“ users’ ”) and, perhaps more importantly, please specify whether this is patient satisfaction, provider satisfaction, or both.

Lines 71-72: this is a sentence fragment

Line 75, 102, 155; Discussion line 92, and others: Suggest the word “refer” instead of “derive”.

Line 82: change “presenting documented” to “presenting with documented”

Line 83: Instead of “were included” should state “were eligible or inclusion” (because some were then excluded).

Line 91: consider using “rectal bleeding” instead of “rectorrhagia”—not because rectorrhagia is incorrect, but it’s an uncommon term in the US.

Line 92: change “cough has” to “cough that has”

Line 94-95: change “significant lymph nodes” to “significantly enlarged lymph nodes”, or perhaps simply “lymphadenopathy”

Line 107: misspellings: “nauseas” and “vomits”

Line 110: misspellings: “glomerular filtrate (GF)” should be “estimated glomerular filtration rate”

Lines 108-118: inconsistent capitalizations – some first letters are capitalized, others are not – should be consistent

Line 108: “erythrocyte” is misspelled. Missing “)” after “Hb”.

Line 107: need comma after “depressive symptoms”

Line 119: suggest “primary” instead of “main” outcome (primary matches with secondary outcome, the term used on line 122).

Line 124: Instead of “Statistical issues” suggest “Statistical Analysis”

Table 1, line 158: suggest “Habits” instead of “Toxic Habits” (“toxic” is a bit judgmental, not used often)

Table 2: “Vomiting” instead of “Vomit”

Line 160: “Patients with psychiatric disorder” – suggest “Patients with psychiatric disorders” or “Patients with a psychiatric disorder”

Table 2: these are not only blood tests, so suggest another title (FOBT are stool tests).

Line 171 “GGT i AP” = ?

Table 3, column heading: “Psichiatric” is misspelled.

Table 3, row heading: “TC” should be “CT”

Table 4: “Placed” should read “Place” or “Location”. Is it “not specified” or actually unknown? “Lax Cardia” is an uncommon term – suggest using a more common term. Some typographic errors (period after “Rectum”, small “i” in Alcohol induced Liver Disease; repeated “Benign” in “Benign Colon Polyps” “Enolism” – is this “Embolism”? “Unknonwn” is a misspelling.

Discussion Line 60, 62: suggest using the term “depression” instead of “depressive syndrome”.

2. Introduction:

Suggest including more information on the specific rationale for this study. For example, were there unanswered questions that previous studies with smaller sample sizes were unable to answer, or were insufficiently powered? Or is the primary aim to determine whether the etiologies of IWL presenting to RDUs is different than those of previous studies in which patients were worked up in a different setting?

3. Methods:

General comments:

-It would be helpful to know whether RDU providers followed an algorithm. Inclusion criteria were that patients could not have symptoms or signs specific to an organ or system, but then providers made decisions on doing a CT scan of the chest, for example, versus the abdomen, or performed upper versus lower endoscopy—presumably there were then some symptoms or exam findings that led to specific diagnostic choices? However, these patients had apparently negative localizing symptoms in order to be included in the study - was the absence of such symptoms based on referral to the RDU, acceptance to the RDU, or the first intake appointment at the RDU?

-Please describe the data extraction – if chart review was required, did multiple investigators review charts for final etiology, and if so, was there agreement (kappa statistic) among reviewers? Or, was all data abstracted from a database?

Line 35: “diagnostic-therapeutic process” – it is uncertain what part of the RDU treatment is therapeutic, as this study appears to be focused on diagnoses of the underlying etiology of IWL. Were there treatments involved?

Lines 73-76: It would be helpful to have more description of entry referral to RDUs. How are patients referred there? Are there referral criteria or acceptance to an RDU? Presumably these are adults, but is there an age criteria (e.g. over 18)? In the study’s setting, do primary care providers not typically order CT scans or refer for endoscopy?

Lines 76-78: Exclusion criteria to the RDU are poorly defined. These should be defined explicitly here or in an appendix of supplementary material. Given that this study hinges on diagnostic etiologies, the exclusion criteria are important: how was “severe functional dependency” defined? For example, was there a rating scale? Were those patients then hospitalized? Why would patients need “family support” to present to an outpatient center? What were the “mild or chronic symptoms”—were these different than the “specific symptoms” starting on line 89?

Line 91: “serious rhythm deposition changes” – this is unclear – do you mean an abnormal EKG/ECG? If so, how abnormal?

Lines 89-99: These exclusion criteria appear to refer to patients who were seen at the RDU, had IWL, but excluded from the analysis. It would be helpful to specify at what point these patients were excluded – one can infer it is based on their initial history and physical examination, presumably at the first RDU visit, but it is not clear. “Malignant hepatomegaly” should be defined – ie is malignancy somehow known at the initial visit? “Subocclusive crises” should also be defined.

Additionally, how is “non-compliance with study criteria for RDU” defined?

What is “death during diagnostic process”? Does this mean the patient died before completing the workup? If so, suggest commenting on why these patients should not be analyzed – for example, if they died of malignancy that would be potentially important data, especially if there were autopsy data and mortality is a secondary outcome measure.

Lines 106-108: Please explain why some gastrointestinal symptoms merit exclusion (I believe this is what “guide symptoms” are? – line 106 –this term should be defined), but others (abdominal pain, nausea, vomiting) are not—is there precedent from previous studies to define these exclusions?

Results:

Figure 1: This figure should be labeled. There are misspellings: “Weight loss <5% o not weight loss,” and “Elegibles”. Terminology is inconsistent – “UWL” is presumably “Unintentional Weight Loss” but “Involuntary” is used elsewhere; “Tumoral” hepatomegaly is used in Figure 1 but “malignant” hepatomegaly is used in the text.

Table 1: Can report only % male or female (both not needed) unless there were significant nonbinary genders that should be reported. “Phychiatric” is a misspelling (column heading). It is unclear what “n valid” means – presumably this is the n for which the subcategory had valid data. “CCEE” is not defined. “UDR” is presumably “RDU”. The table results should be discussed in the text. For example, there is a p=0.001 for the referral sources, but the clinical significance is not discussed—presumably the exact test was used given the low n in the Unknown origin column and the endoscopic procedures row- however is the intent to show that there is a higher relative percentage of primary healthcare center referrals in the unknown origin group compared to the others?

The onset of weight loss is presented as significantly difference (p=0.018), presumably chi-squared or exact test given low n in the Unknown Origin group, but there should be discussion in the text regarding the clinical significance – the test statistic only suggests that the patients are not evenly spread among these groups, but there is no obvious trend (e.g. is the > 6 months trend significant? Do the authors feel that it is reasonable to say that the Unknown group had a higher percentage of patients who had > 6 months of IWL? Ie what is the take-home point, if any, of this part of the table?)

Table 2: In methods it was stated that quantitative variables would be reported as mean and SD—however, these variables’ data types are not reported in the Table. It should clearly state mean (SD), or more likely median (IQR) or mean (95%CI) based on the format showing a range. The “FOBT” row should be defined (is this that FOBT was done vs not done?).

Table 3: Line 192—it is unclear where the 6.9% figure comes from.

Figure 2: suggest the line should be “Mortality” and not “Mortality function”

Discussion:

Good summary of overall etiologies and literature.

Discussion mainly talks about imaging. There are other interesting findings that deserve discussion and context with regard to prior literature as well as clinical significance: for example--

-Lab abnormalities: many were statistically significant, but only some appeared to be different enough to be potentially clinically useful. What do the authors think?

-It is interesting that the mean weight loss was not different between groups – is this surprising? How does it compare to previous studies?

-Alcohol consumption in Table 1 is shows as being in a significantly greater percentage of patients in the Neoplastic group, but the quantified grams per day was not significant. How do the authors interpret this discrepancy? Is it similar to prior studies?

Discussion should include limitations; could there be variability between providers at the RDU in how they conduct the workup? Consider discussing the limitations of the exclusion criteria, including dementia, lack of “family support”,etc, as it affects generalizability.

Given the sheer number of statistical tests performed, consider discussing whether the study has a higher chance of type I errors.

Line 46-48 – pls clarify – I believe you mean that the differences are due to including hospitalized patients in the other studies, eg “These differences may be due to … with specific symptoms in the other previously published studies.”

Line 61 “The two studies” – pls state which two studies you are referring to (Bilbao-Garay and the authors’ study?).

Lines 89-92: interesting idea, but could use more justification – are primary care providers not allowed to order CT scans or refer patients for endoscopy? It would help to understand the setting better in terms of generalizability. Similar to lines 66-67--there was endoscopy and CT scanning 10-20 years ago – at least in the US there was no lack of these modalities.

Suggest using “yield” instead of “profitable” – the latter implies cost, while here we are really interested in diagnostic yield.

References: 5 and 6 are duplicates.

Reviewer #2: In this paper the authors reviewed retrospectively patients presented to the rapid diagnostic unit (RDU) for isolated involuntary weight loss (IWL) in a period between 2005 and 2013. The study was a single-center one. Follow up was performed at 1 year. From the 1592 identified patients, 791 (49.7%) met the criteria for enrollment into the study, which is the largest study to date for this group of patients. The main results of the study are that non-malignant organic diseases was the cause for IWL in 44.5%, psychiatric disorders in 29.0%, malignant diseases in 23.6%, and unknown causes in 3.2%. Mortality at 12 months was 18.6% overall, which was much higher in the malignant group 61.1% (95%CI: 54.2-68.2). The authors conclude that malignancy should be ruled out during the first visit for patients presenting with IWL.

The manuscript is pretty well written, graphs illustrative, and its findings could be of some interest. The results of the paper are however not novel, add little to the existing data, but the cohort is large, though retrospective. The discussion section could have been constructed in a more attractive way.

Besides, I think there are some issues with this paper:

Major:

1. The main drawbacks are that it is a single center study with retrospective design. Also, the studied period ended with more than 7 years ago, which is of some concern. Why wasn’t the study period ended sooner, such as in 2019, with follow-up ending in 2020?

2. Materials and methods: the authors state that the study included “patients visited at the RDU between January 2005 and December 2013”, and that the “follow-up of the last patient was done in December 2014”. However, below, the authors state that the follow-up was of 6 months, which is different than the 1-year interval presented above. Please explain the difference.

3. Materials and methods: Please explain why patients with mild or chronic symptoms represented exclusion criteria for being referred to the RDU, because it is not clear, and because I suppose that most of the enrolled patients had some mild and chronic symptoms.

4. Results: Please review the fact that “mean weight loss in the past 12 months was 8.3 Kg (SD 4.7)”, because this would mean that some of the patients lost less than 5kg, which is a non-diagnostic criterion for IWL. See also the data in table 1, in which SD seems to be discordant between the columns of the amounts of weight loss.

5. In the conclusion of the paper, the authors state that “computed tomography is the most profitable complementary test to be performed in addition to complete anamnesis and blood tests for a good diagnostic approximation.” However, there is no discussion throughout the paper about the value of complete anamnesis and of blood tests in establishing the diagnosis in this cohort of patients.

Minor:

1. Abstract: please replace “Mortality at 12 months was higher…” with “Mortality at 12 months was much higher…”, as the differences are quite important between the two groups.

2. Introduction: please replace “ambulatory rapid diagnostic unit (RDU) allow us…” by “ambulatory rapid diagnostic units (RDUs) allow us…”. Also, “In these RDU…” by “In these RDUs…”.

3. Introduction: please review the expression “organic or malignant 32 to 51%”; it could be organic or non-malignant, and there could be a mistake.

4. Material and methods: “Ethical Committee number CEIC 15/16” should be integrated into a sentence.

5. Material and methods: it is not clear what it is meant by “serious rhythm deposition changes”. Please rephrase or explain better.

6. Material and methods: The comma after “lymph nodes” should be deleted.

7. Table 1: change “UDR” to “RDU”.

8. Table 2: please rephrase “ESR, GGT i AP”.

9. Table 2: if the values between square brackets represent percentile 25-percentile 75, please state as such: [percentile 25-percentile 75], and not “(percentile 25-percentil75)”.

10. Results: please rephrase “11.9% of thoracoabdominal CTs showed images suspicious for malignancy (which proved to be false positive…” to “11.9% of thoracoabdominal CTs showed images considered suspicious for malignancy (which proved to be false positive…”.

11. Results: please add results of performing upper gastrointestinal endoscopy, not only colonoscopy, especially as below the authors state that “In the neoplastic group, 50% of gastroscopies… were histologically confirmed as malignant.”

12. Table 4: delete repeated word (benign). What do you mean by significant when referring to colon polyps?

13. Table 4: what does enolism stand for?

14. Results: please put “per cent” together.

15. Discussion: please rephrase: “Computed tomography is the most profitable complementary test”; profitable does not seem appropriate for this instance.

16. Figure 3: what do the figures below the graph represent (no. of patients at risk)? Is it the number of patients followed-up at the specific time intervals? It is not clear.

6. PLOS authors have the option to publish the peer review history of their article (what does this mean?). If published, this will include your full peer review and any attached files.

Reviewer #1: No

Reviewer #2: No

---

## [Author Response · Author response to Decision Letter 0]

26 Mar 2021

Dear Editor and Reviewers, 

The authors are grateful for your prompt review of the draft manuscript PONE-D-20-05824. Your comments are highly appreciated and have helped us to introduce important improvements in our paper.

Please find below a point-by-point response to all the reviewers’ comments, along with the amended version of the document with the new text presented in red, which we are resubmitting for consideration and possible publication.

Yours faithfully,

Jordi Aligué, MD, PhD

Althaia Xarxa Assistencial Universitària de Manresa

Point by point response to the reviewers’ comments

Ref: PONE-D-20-05824 

“Etiologies and mortality at 12 months of patients with isolated involuntary weight loss in a rapid diagnostic unit”

Reviewer #1: Thank you for the opportunity to review this interesting study.

Thank you for your careful review of the manuscript, which has enabled us to introduce some significant improvements.

MINOR CHANGES: 

Following your recommendations we have incorporated all the minor changes.

Specific minor comment #1

1. I suggest having a native English speaker review content for grammar, style, and spell check.

Response: A native English-speaking scientific editor has revised the new version of the manuscript for grammar, style and spelling.

Examples:

Specific minor comment #1.1

Line 32: change “visited at” to “presenting to”

Response: Modified 

Specific minor comment #1.2

Line 34: spell out “computed tomography” for the first time “CT” is used.

Response: Modified 

 

Specific minor comment #1.3

Line 40: “… which was higher in the malignant group” is awkward phrasing – suggest “…with the mortality rate being the highest in those with malignancy (61.1%, 95% CI 54.2,68.2)

Response: Modified 

Specific minor comment #1.4

Line 57: pluralize “rapid diagnostic unit”, or, if singular, then on line 56, add “an” before “ambulatory” 

Response: Modified

Specific minor comment #1.5

Line 58: “users”: this should be possessive (“ users’ ”) and, perhaps more importantly, please specify whether this is patient satisfaction, provider satisfaction, or both.

Response: Modified 

Specific minor comment #1.6

Lines 71-72: this is a sentence fragment

Response: Modified 

Specific minor comment #1.7

Line 75, 102, 155; Discussion line 92, and others: Suggest the word “refer” instead of “derive”.

Response: Modified

 

Specific minor comment #1.8

Line 82: change “presenting documented” to “presenting with documented”

Response: Modified

Specific minor comment #1.9

Line 83: Instead of “were included” should state “were eligible or inclusion” (because some were then excluded).

Response: Modified

Specific minor comment #1.10

Line 91: consider using “rectal bleeding” instead of “rectorrhagia”—not because rectorrhagia is incorrect, but it’s an uncommon term in the US.

Response: Modified

Specific minor comment #1.11

Line 92: change “cough has” to “cough that has”

Response: Modified

Specific minor comment #1.12

Line 94-95: change “significant lymph nodes” to “significantly enlarged lymph nodes”, or perhaps simply “lymphadenopathy”

Response: Modified

Specific minor comment #1.13

Line 107: misspellings: “nauseas” and “vomits”

Response: Modified

Specific minor comment #1.14

Line 110: misspellings: “glomerular filtrate (GF)” should be “estimated glomerular filtration rate”

Response: Modified

Specific minor comment #1.15

Lines 108-118: inconsistent capitalizations – some first letters are capitalized, others are not – should be consistent

Response: Modified

Specific minor comment #1.16

Line 108: “erythrocyte” is misspelled. Missing “)” after “Hb”.

Response: Modified

Specific minor comment #1.17

Line 107: need comma after “depressive symptoms”

Response: Modified

Specific minor comment #1.18

Line 119: suggest “primary” instead of “main” outcome (primary matches with secondary outcome, the term used on line 122).

Response: Modified

Specific minor comment #1.19

Line 124: Instead of “Statistical issues” suggest “Statistical Analysis”

Response: Modified

Specific minor comment #1.20

Table 1, line 158: suggest “Habits” instead of “Toxic Habits” (“toxic” is a bit judgmental, not used often)

Response: Modified

Specific minor comment #1.21

Table 2: “Vomiting” instead of “Vomit”

Response: Modified

Specific minor comment #1.22

Line 160: “Patients with psychiatric disorder” – suggest “Patients with psychiatric disorders” or “Patients with a psychiatric disorder”

Response: Modified

Specific minor comment #1.23

Table 2: these are not only blood tests, so suggest another title (FOBT are stool tests).

Response: In response to this comment we have changed the heading of table 2, as follows:

“Table 2. Blood and stool analysis parameters of patients with IWL, overall and according to etiology”

Specific minor comment #1.24

Line 171 “GGT i AP” = ?

Response: We have changed AP to ALP, which stands for alkaline phosphatase (ALP).

 

Specific minor comment #1.25

Table 3, column heading: “Psichiatric” is misspelled.

Response: Modified

Specific minor comment #1.26

Table 3, row heading: “TC” should be “CT”

Response: Modified

Specific minor comment #1.27

Table 4: “Placed” should read “Place” or “Location”. Is it “not specified” or actually unknown? 

Response: We have changed “Placed not specified” to “Unknown primary site”

Specific minor comment #1.28

“Lax Cardia” is an uncommon term – suggest using a more common term. 

Response: We have changed “Lax Cardia” to “Incompetent cardia”

Specific minor comment #1.29

Some typographic errors (period after “Rectum”, small “i” in Alcohol induced Liver 

Response: Modified

Specific minor comment #1.30

Disease; repeated “Benign” in “Benign Colon Polyps” 

Response: Modified

Specific minor comment #1.31

“Enolism” – is this “Embolism”? 

Response: The reference is to excessive alcohol consumption. We have changed “Acive enolism” to “excessive alcohol consumption” (≥30g/day in women and ≥40g/day in men)”.

Specific minor comment #1.32

“Unknonwn” is a misspelling.

Response: Modified

Specific minor comment #1.33

Discussion Line 60, 62: suggest using the term “depression” instead of “depressive syndrome”.

Response: Modified

MAJOR CHANGES:

Introduction:

Specific major comment #1

Suggest including more information on the specific rationale for this study. For example, were there unanswered questions that previous studies with smaller sample sizes were unable to answer, or were insufficiently powered? Or is the primary aim to determine whether the etiologies of IWL presenting to RDUs is different than those of previous studies in which patients were worked up in a different setting?

Response: We have added the following paragraph in the introduction: 

“Most studies published on IWL include small samples of patients with specific symptoms, which may help to guide further investigations in particular areas. Few studies have focused on isolated weight loss in larger samples [3,4,6–10]. What is more, studies on patients with weight loss are traditionally performed in hospitalized patients, but today, outpatient rapid diagnostic units (RDUs) allow us to optimize resources and achieve high levels of satisfaction among both patients and providers [11,12]. In these RDUs, the most frequently diagnosed pathology is cancer (18-30%) [13,14]. Few studies have been carried out entirely in patients with isolated IWL treated in RDUs, who may present different etiologies from those described in studies performed in outpatients evaluated using conventional means or in hospitalized patients.”

Methods:

Specific major comment #2

General comments:

-It would be helpful to know whether RDU providers followed an algorithm. Inclusion criteria were that patients could not have symptoms or signs specific to an organ or system, but then providers made decisions on doing a CT scan of the chest, for example, versus the abdomen, or performed upper versus lower endoscopy—presumably there were then some symptoms or exam findings that led to specific diagnostic choices? However, these patients had apparently negative localizing symptoms in order to be included in the study - was the absence of such symptoms based on referral to the RDU, acceptance to the RDU, or the first intake appointment at the RDU?

Response: To clarify the RDU referral algorithm, we have added a table in the supplementary materials detailing the signs and symptoms that warrant referral to the internal medicine RDUs. 

Specific major comment #3

-Please describe the data extraction – if chart review was required, did multiple investigators review charts for final etiology, and if so, was there agreement (kappa statistic) among reviewers? Or, was all data abstracted from a database?. 

Response: The final diagnosis was made by consensus between two physicians at the internal medicine service. We have modified the corresponding paragraph in the Statistical analysis as follows:

“The primary outcome was etiology of IWL, classified as neoplasia, non-malignant organic disease, psychiatric or unknown (i.e., when not determined after 12 months of follow-up). The final diagnosis was made by consensus between two physicians at the internal medicine service. The secondary outcome measure was one-year mortality.”

Specific major comment #4

Line 35: “diagnostic-therapeutic process” – it is uncertain what part of the RDU treatment is therapeutic, as this study appears to be focused on diagnoses of the underlying etiology of IWL. Were there treatments involved?

Response: We have changed “diagnostic-therapeutic” process to “diagnostic process”, since the study focuses only on diagnoses. 

“Methods: A single-center retrospective observational study including all patients presenting to the RDU for isolated IWL between 2005 and 2013. The authors collected demographic and clinical variables, results of complementary tests (blood tests, x-rays, computed tomography CT scan and digestive endoscopy), main diagnosis and vital status at 12 months”.

 

Specific major comment #5

Lines 73-76: It would be helpful to have more description of entry referral to RDUs. How are patients referred there? Are there referral criteria or acceptance to an RDU? Presumably these are adults, but is there an age criteria (e.g. over 18)? In the study’s setting, do primary care providers not typically order CT scans or refer for endoscopy?

Response: 

As noted above in specific major comment #2, we have added a table as supplementary material detailing the signs and symptoms warranting referral to internal medicine RDUs.

Primary care physicians may request additional tests such as CT scans or digestive endocopies, but if there is a high suspicion of malignancy, the patient is referred to the RDU in order to obtain the diagnosis within the recommended period of 1 month.

Specific major comment #6

Lines 76-78: Exclusion criteria to the RDU are poorly defined. These should be defined explicitly here or in an appendix of supplementary material. Given that this study hinges on diagnostic etiologies, the exclusion criteria are important: how was “severe functional dependency” defined? For example, was there a rating scale? Were those patients then hospitalized? Why would patients need “family support” to present to an outpatient center? What were the “mild or chronic symptoms”—were these different than the “specific symptoms” starting on line 89?

Response: 

Following the reviewer’s recommendations, we now define each of the exclusion criteria more clearly. Functional dependency was assessed with the Barthel index, cognitive impairment with the Global Deterioration Scale, and symptoms of more than 12 months of evolution were considered to be chronic. Dependent patients or those who did not have family able to accompany them on RDU outpatient visits were hospitalized.

“Exclusion criteria for RDU were severe functional dependency (Barthel Index≤20), cognitive decline (Global Deterioration Scale>3), lack of family support for attending the outpatient center and chronic symptoms (more than 12 months of evolution)“.

 

Specific major comment #7

Line 91: “serious rhythm deposition changes” – this is unclear – do you mean an abnormal EKG/ECG? If so, how abnormal?

Response: In the interests of clarity, we have changed “serious rhythm deposition changes” to “intestinal transit disorders”.

Specific major comment #8

Lines 89-99: These exclusion criteria appear to refer to patients who were seen at the RDU, had IWL, but excluded from the analysis. It would be helpful to specify at what point these patients were excluded – one can infer it is based on their initial history and physical examination, presumably at the first RDU visit, but it is not clear. 

Response: As the reviewer suggests, we now specify that the exclusion criteria were checked at the first RDU visit. All patients who did not meet the criteria for isolated IWL were excluded from the study.

Specific major comment #9

“Malignant hepatomegaly” should be defined – ie is malignancy somehow known at the initial visit? 

Response: We have replaced “malignant hepatomegaly” with “suspicion of malignant hepatomegaly according to physical examination or diagnostic imaging previous to the initial RDU visit”. 

Specific major comment #10

“Subocclusive crises” should also be defined.

Additionally, how is “non-compliance with study criteria for RDU” defined?

Response: To clarify the concept of “Subocclusive crises” we now use the term “intestinal subocclusion crises”. We have also replaced “non-compliance with study criteria for RDU” to “non-compliance with referral criteria for RDU”. 

 

Specific major comment #11

What is “death during diagnostic process”? Does this mean the patient died before completing the workup? If so, suggest commenting on why these patients should not be analyzed – for example, if they died of malignancy that would be potentially important data, especially if there were autopsy data and mortality is a secondary outcome measure.

Response: Yes, the idea is that the patient died during the diagnostic process, and the diagnostic tests to determine the etiology could not be performed. Nor was an autopsy performed to rule out malignancy. To clarify this point we have expanded our description of the concept of “death during diagnostic process” as follows:

“Patients who died during the diagnostic process and in whom it was not able to perform the required diagnostic tests were also excluded”.

Specific major comment #12

Lines 106-108: Please explain why some gastrointestinal symptoms merit exclusion (I believe this is what “guide symptoms” are? – line 106 –this term should be defined), but others (abdominal pain, nausea, vomiting) are not—is there precedent from previous studies to define these exclusions?

Response: Effectively, non-specific gastrointestinal symptoms not considered as referral criteria to RDU, such as abdominal pain or nausea/vomiting, were not considered exclusion criteria for isolated IWL.

To clarify this point we have modified this paragraph as follows:

“associated unspecific symptoms (asthenia, anorexia, fever, depressive symptoms, abdominal pain, nausea and/or vomiting, arthralgias and benign lymph nodes) which were not considered as symptoms warranting referral”.

We have changed the term “guide symptoms” to “symptoms warranting referral”

Results:

Specific major comment #13

Figure 1: This figure should be labeled. There are misspellings: “Weight loss <5% o not weight loss,” and “Elegibles”. Terminology is inconsistent – “UWL” is presumably “Unintentional Weight Loss” but “Involuntary” is used elsewhere; “Tumoral” hepatomegaly is used in Figure 1 but “malignant” hepatomegaly is used in the text.

Response: We have introduced all the proposed changes.

Specific major comment #14

Table 1: Can report only % male or female (both not needed) unless there were significant nonbinary genders that should be reported. 

Modified

“Phychiatric” is a misspelling (column heading). It is unclear what “n valid” means – presumably this is the n for which the subcategory had valid data. “CCEE” is not defined.

Modified

 “UDR” is presumably “RDU”. 

Modified

The table results should be discussed in the text. For example, there is a p=0.001 for the referral sources, but the clinical significance is not discussed—presumably the exact test was used given the low n in the Unknown origin column and the endoscopic procedures row- however is the intent to show that there is a higher relative percentage of primary healthcare center referrals in the unknown origin group compared to the others?

Response: We now explain this point in the paragraph in Results that discusses table 1. 

“Table 1 shows patients’ baseline characteristics. Mean age was 67.9 years (SD 15.7), and 50.4% were men. Statistically significant differences (p<0.001) were observed in the distribution of sexes between the four diagnostic groups: the highest percentage of men was found in the neoplastic group (67.0%) and the highest presence of women in the psychiatric disorder group (65.1%). Most patients were referred from primary care (69.2%). In the group of patients with IWL of unknown origin, there was a higher percentage referred from primary healthcare centers. In the group of neoplasic patients, the percentage referred from the radiology department, and who had undergone endoscopic procedures, was greater (p=0.001)”.

Specific major comment #15

The onset of weight loss is presented as significantly difference (p=0.018), presumably chi-squared or exact test given low n in the Unknown Origin group, but there should be discussion in the text regarding the clinical significance – the test statistic only suggests that the patients are not evenly spread among these groups, but there is no obvious trend (e.g. is the > 6 months trend significant? Do the authors feel that it is reasonable to say that the Unknown group had a higher percentage of patients who had > 6 months of IWL? Ie what is the take-home point, if any, of this part of the table?)

Response: We now explain this point in the paragraph in Results that discusses table 1. 

“In the cancer and non-malignant organic disease groups, almost 50% had had IWL for less than three months, while in psychiatric disorders and unknown origin groups, most had had IWL for more than six months.”

Specific major comment #16

Table 2: In methods it was stated that quantitative variables would be reported as mean and SD—however, these variables’ data types are not reported in the Table. It should clearly state mean (SD), or more likely median (IQR) or mean (95%CI) based on the format showing a range. The “FOBT” row should be defined (is this that FOBT was done vs not done?).

Response: The reviewer is right. Following his/her recommendations, in the “Statistical analysis” section we summarize the continuous variables that did not have a normal distribution.

“Categorical variables are presented as absolute values and relative frequencies. Continuous variables are summarized as means and standard deviations (SD) for normal distributions and as medians and interquartile range [IQR] for non-normal distributions. For the bivariate analysis the one-way ANOVA was used for continuous variables. Either the chi-square test, Fisher’s exact test or the Monte Carlo method was used for categorical variables.”

As regards ”FOBT”, yes, we mean FOBT done/not done. We have now replaced the heading “FOBT” with “FOBT (Yes)”.

Specific major comment #17

Table 3: Line 192—it is unclear where the 6.9% figure comes from.

Response: We have modified the text as follows.

“Of the 72 X-rays performed at other sites, 6.9% were suspicious for malignancy. This percentage was statistically higher in the neoplastic group (26.3%; p=0.039).”

Specific major comment #18

Figure 2: suggest the line should be “Mortality” and not “Mortality function”

Response: modified.

Discussion:

Good summary of overall etiologies and literature.

Specific major comment #19

Discussion mainly talks about imaging. There are other interesting findings that deserve discussion and context with regard to prior literature as well as clinical significance: for example, Lab abnormalities: many were statistically significant, but only some appeared to be different enough to be potentially clinically useful. What do the authors think?.

Response: The reviewer is right; we had not discussed the clinical implications of the abnormal laboratory parameters in the discussion. We have now added the following paragraph on this issue:

“The presence of anemia, low serum albumin, iron deficiency and higher levels of leukocytes, ESR, GGT, ALP, ferritin, TSH, LDH and CRP was associated with an increased risk of cancer, as in the study by Baicus et al. Therefore, any alteration of these basic analytical parameters in patients presenting with isolated IWL raises the suspicion of inflammatory processes, localized neoplastic processes, or disseminated disease. For example, ferritin may be increased in different types of cancer, GGT in cases of liver metastasis and ALP in cases of liver and/or bone metastases.”

 

Specific major comment #20

-It is interesting that the mean weight loss was not different between groups – is this surprising? How does it compare to previous studies?

Response: On this point there is a certain amount of disagreement in the literature. Some studies report differences in mean weight loss according to the etiology of isolated IWL but others do not observe significant differences between the groups (as in our case). We have added the following paragraph on this issue:

“Unlike other studies [3,7,15–18,26], we found a lower percentage of IWL of unknown cause (3.2%). This may be due to the greater availability of complementary tests (both imaging tests and endoscopy), the long follow-up to confirm diagnosis, and the appraisal of the patient’s psychosocial situation (an aspect not included in most studies). 

In agreement with Rabinovitz et al. and Metalidis et al., we did not find significant differences in degree of weight loss with respect to etiology. In contrast, Bosch et al. reported that cancer patients had more pronounced weight loss than other patients. However, these studies included patients not only with isolated IWL but also IWL as a dominant feature of disease.”

Specific major comment #21

-Alcohol consumption in Table 1 is shows as being in a significantly greater percentage of patients in the Neoplastic group, but the quantified grams per day was not significant. How do the authors interpret this discrepancy? Is it similar to prior studies?

Response: 

Indeed, the group of patients with neoplasia included a higher percentage of drinkers. However, no significant differences were observed in the average grams of alcohol consumed per day between drinkers in the different groups. The patients who consumed the most alcohol per day were the ones with psychiatric disorders and the ones with non-malignant organic pathology. In the psychiatric pathology group there may have been a higher consumption of alcohol among drinkers due to the underlying pathology (85.6% had depression). Additionally, the group of patients with non- malignant organic disease included patients with chronic alcohol-induced liver disease or pancreatitis.

 

Specific major comment #22

Discussion should include limitations; could there be variability between providers at the RDU in how they conduct the workup? Consider discussing the limitations of the exclusion criteria, including dementia, lack of “family support”,etc, as it affects generalizability.

Response: 

At the suggestion of the reviewer we have modified and extended the paragraph on the study’s limitations in the Discussion.

“Our study has some limitations. First, no studies are available on isolated IWL performed exclusively in primary care, where non-malignant organic disease and psychiatric disorder are likely to be more prevalent causes than in our sample. Our study was carried out in a RDU; therefore, many patients with non-malignant organic disease and psychiatric disorder may have previously been diagnosed and treated in primary care, and would not have been referred to the RDU. Although Primary Care physicians may request complementary tests such as imaging tests or endoscopies, in our setting, when there is a suspicion of malignancy, patients are referred to the RDU. Consequently, pathologies such as diabetes mellitus, hyperthyroidism or depression which can be diagnosed in primary care with a standard clinical assessment and analysis, may be more frequent in the general population with IWL than in our sample. In addition, patients with dementia or severe dependency or lack of family support could be diagnosed with a malignant disorder outside the RDU. Second, the inclusion and definition criteria used to refer patients to RDU, and the procedures for conducting workup, may differ substantially between providers. This means that epidemiological and demographic characteristics of evaluable patients may differ between hospitals. Another source of variability may be differences in the ability of the physicians evaluating IWL. As a result, the implications of our study cannot be directly generalized to other settings. However, other studies have found similar results in the prevalence of cancer among patients with IWL attended at RDUs. Third, 4.7% of patients were excluded without obtaining a diagnosis, either because the patient refused follow-up at the RDU or because complementary tests were not performed.”

Specific major comment #23

Given the sheer number of statistical tests performed, consider discussing whether the study has a higher chance of type I errors.

Response: We agree with the reviewer that due to the high number of statistical tests performed, it is possible to have detected a statistically significant difference when in fact it did not exist (type I error). 

Specific major comment #24

Line 46-48 – pls clarify – I believe you mean that the differences are due to including hospitalized patients in the other studies, eg “These differences may be due to … with specific symptoms in the other previously published studies.”

Response: we have modified the second paragaph of the Discussion to highlight the reasons for the differences in the prevalence of cancer in the various studies.

“In our study, the prevalence of cancer in patients with involuntary weight loss was similar to that recorded in some previous studies (22-24.8%) [3,15,19,22], but lower than those found in others (33-36.3%) [16,17,23]. These differences may be due to the inclusion of hospitalized patients in the other studies [17,23] (evaluated outside the RDU), or in the case of studies of IWL performed in RDUs, the inclusion of patients with other specific symptoms [16]. Consistent with other published series, malignancies were more frequent in men [10,16,24] and in older patients [3,10,16,24].” 

Specific major comment #25

Line 61 “The two studies” – pls state which two studies you are referring to (Bilbao-Garay and the authors’ study?).

Response: We have modified the sentence to clarify this point.

“In our study the prevalence of depressive syndrome and anxiety in the psychiatric disorders group was around 94%, similar to the rate recorded by Bilbao-Garay et al.”

 

Specific major comment #26

Lines 89-92: interesting idea, but could use more justification – are primary care providers not allowed to order CT scans or refer patients for endoscopy? It would help to understand the setting better in terms of generalizability. Similar to lines 66-67--there was endoscopy and CT scanning 10-20 years ago – at least in the US there was no lack of these modalities.

Response: Although Primary Care physicians may request complementary tests such as imaging tests or endoscopies, in our setting, when there is a suspicion of malignancy, patients are referred to the RDU in order to obtain the diagnosis within the recommended time period of less than 30 days.

We now mention this point in the limitations section of the study:

“Although Primary Care physicians may request complementary tests such as imaging tests or endoscopies, in our setting, when there is a suspicion of malignancy, patients are referred to the RDU”.

It is true that CT scanning was available 10-20 years ago, but the fact is that in most studies performed in that period, the percentage of patients who underwent CT scanning was low. Nevertheless, we have removed the following sentence from the text: “compared to previous studies performed 10-20 years ago”.

“Compared to other studies [3,7,15–18,26], we found a lower percentage of IWL of unknown cause (3.2%). This may be due to the greater availability of complementary tests (e.g., imaging tests and endoscopy), the long follow-up to confirm diagnosis, and the appraisal of the patient’s psychosocial situation (an aspect not included in most studies).”

Specific major comment #27

Suggest using “yield” instead of “profitable” – the latter implies cost, while here we are really interested in diagnostic yield.

Response: Modified

 

Specific major comment #28

References: 5 and 6 are duplicates.

Response: 

We have merged the two references, and in response to the reviewers’ suggestions we have added some bibliographical references that did not appear in the previous version. 

Reviewer #2: 

In this paper the authors reviewed retrospectively patients presented to the rapid diagnostic unit (RDU) for isolated involuntary weight loss (IWL) in a period between 2005 and 2013. The study was a single-center one. Follow up was performed at 1 year. From the 1592 identified patients, 791 (49.7%) met the criteria for enrollment into the study, which is the largest study to date for this group of patients. The main results of the study are that non-malignant organic diseases was the cause for IWL in 44.5%, psychiatric disorders in 29.0%, malignant diseases in 23.6%, and unknown causes in 3.2%. Mortality at 12 months was 18.6% overall, which was much higher in the malignant group 61.1% (95%CI: 54.2-68.2). The authors conclude that malignancy should be ruled out during the first visit for patients presenting with IWL.

The manuscript is pretty well written, graphs illustrative, and its findings could be of some interest. The results of the paper are however not novel, add little to the existing data, but the cohort is large, though retrospective. The discussion section could have been constructed in a more attractive way.

In response to the reviewer’s suggestion, we have substantially modified the discussion and now expand our interpretation of the results.

Besides, I think there are some issues with this paper:

MAJOR CHANGES: 

Specific major comment #1

The main drawbacks are that it is a single center study with retrospective design. Also, the studied period ended with more than 7 years ago, which is of some concern. Why wasn’t the study period ended sooner, such as in 2019, with follow-up ending in 2020?

Response: 

This study was carried out as part of a PhD thesis carried out by an internal medicine physician, who began work on the thesis in 2014. The thesis was presented in 2018, and the writing of the manuscript began after this date. The main author is currently working on the derivation of a predictive model to rule out malignancy in patients with isolated IWL based on their demographic, clinical and analytical parameters. This model will be validated using the data of the patients treated at the RDU in the period between 2016 and 2018 with the end of the follow-up in 2019.

 

Materials and methods:

Specific major comment #2

The authors state that the study included “patients visited at the RDU between January 2005 and December 2013”, and that the “follow-up of the last patient was done in December 2014”. However, below, the authors state that the follow-up was of 6 months, which is different than the 1-year interval presented above. Please explain the difference.

Response: We have modified this paragraph as follows:

“The primary outcome was the etiology of IWL, grouped as neoplasia, non-malignant organic disease, psychiatric and isolated and unknown origin (i.e., when the etiology was not determined after 12-month follow-up).”

All patients were followed up for 12 months. If, after this period of time the etiology of IWL could not be determined, it was recorded as “unknown origin”. 

Specific major comment #3

Please explain why patients with mild or chronic symptoms represented exclusion criteria for being referred to the RDU, because it is not clear, and because I suppose that most of the enrolled patients had some mild and chronic symptoms.

Response: Following the reviewer's recommendations and to clarify this point, we have added a table in the supplementary material where we specify the signs and symptoms that warrant referral to the RDU. The rest of the signs and symptoms are considered as associated symptoms, and are described in table 1. 

 

Results: 

Specific major comment #4

Please review the fact that “mean weight loss in the past 12 months was 8.3 Kg (SD 4.7)”, because this would mean that some of the patients lost less than 5kg, which is a non-diagnostic criterion for IWL. See also the data in table 1, in which SD seems to be discordant between the columns of the amounts of weight loss.

Response: The criterion used for IWL was: 

“All patients presenting with documented isolated IWL of at least 5% over the past 12 months were eligible for inclusion. IWL was considered isolated when it was not accompanied by symptoms or signs specific of a particular organ or system. When weight loss was not documented, the criteria of Marton et al. were used: i.e., patients were eligible if they met at least two of the following criteria: evidence of change in clothes size, confirmation of weight loss by a friend or relative, and ability to give a numerical estimate of weight loss.”

In some patients a weight loss of 5% was lower than 5 kg.

 

Conclusion:

Specific major comment #5

In the conclusion of the paper, the authors state that “computed tomography is the most profitable complementary test to be performed in addition to complete anamnesis and blood tests for a good diagnostic approximation.” However, there is no discussion throughout the paper about the value of complete anamnesis and of blood tests in establishing the diagnosis in this cohort of patients.

Response: Following the recommendation of the reviewer, we have added a paragraph in the discussion highlighting the value of the anamnesis and blood tests for diagnosing isolated IWL.

“A careful history may be very useful for identifying signs or symptoms that may guide further investigations. However, there is no consensus on the tests that should be included in the initial evaluation of isolated IWL. Nonetheless, most authors seem to agree that a detailed medical history, a thorough physical examination, a complete blood test and chest X-ray seem to be sufficient for an initial evaluation.

The presence of anemia, low serum albumin, iron deficiency and higher levels of leukocytes, ESR, GGT, ALP, ferritin, TSH, LDH and CRP was associated with an increased risk of cancer, as in the study by Baicus et al. Therefore, any alteration of these basic analytical parameters in patients presenting with isolated IWL raise the suspicion of inflammatory processes, localized neoplastic processes, or disseminated disease. For example, ferritin may be increased in different types of cancer, GGT in cases of liver metastasis and ALP in cases of liver and/or bone metastases.”

MINOR CHANGES: 

We have introduced all the minor changes recommended by the reviewer.

Abstract:

Specific minor comment #1

Please replace “Mortality at 12 months was higher…” with “Mortality at 12 months was much higher…”, as the differences are quite important between the two groups.

Response: In response to the recommendations of both reviewers, we have modified the sentence as follows:

“Overall mortality at 12 months was 18.6% (95%CI: 16.1-21.6); the highest rate was recorded in the malignancy group (61.1%; 95%CI: 54.2-68.2)”.

Introduction:

Specific minor comment #2

Please replace “ambulatory rapid diagnostic unit (RDU) allow us…” by “ambulatory rapid diagnostic units (RDUs) allow us…”. Also, “In these RDU…” by “In these RDUs…”.

Response: Modified

Specific minor comment #3

Please review the expression “organic or malignant 32 to 51%”; it could be organic or non-malignant, and there could be a mistake.

Response: Modified

Material and methods:

Specific minor comment #4

“Ethical Committee number CEIC 15/16” should be integrated into a sentence.

Response: Modified

Specific minor comment #5

It is not clear what it is meant by “serious rhythm deposition changes”. Please rephrase or explain better.

Response: We have changed “serious rhythm deposition changes” to “intestinal transit disorders”

Specific minor comment #6

The comma after “lymph nodes” should be deleted.

Response: Modified

Specific minor comment #7

Table 1: change “UDR” to “RDU”.

Response: Modified

Specific minor comment #8

Table 2: please rephrase “ESR, GGT i AP”.

Response: Modified

Specific minor comment #9

Table 2: if the values between square brackets represent percentile 25-percentile 75, please state as such: [percentile 25-percentile 75], and not “(percentile 25-percentil75)”.

Response: Modified 

 

Results:

Specific minor comment #10

Please rephrase “11.9% of thoracoabdominal CTs showed images suspicious for malignancy (which proved to be false positive…” to “11.9% of thoracoabdominal CTs showed images considered suspicious for malignancy (which proved to be false positive…”.

Response: Modified 

Specific minor comment #11

Please add results of performing upper gastrointestinal endoscopy, not only colonoscopy, especially as below the authors state that “In the neoplastic group, 50% of gastroscopies… were histologically confirmed as malignant.”

Response: We have modified this paragraph as follows:

“Gastroscopy was performed in 272 patients (34.4%) and colonoscopy in 122 (15.4%). Malignancy was histologically confirmed in 39 gastroscopies (14.3%) and 22 colonoscopies (18%). In the neoplastic group, 50% of gastroscopies and 53.7% of colonoscopies were histologically confirmed as malignant.”

Specific minor comment #12

Table 4: delete repeated word (benign). What do you mean by significant when referring to colon polyps?

Response: Modified. We have rephrased the reference to colon polyps as follows:

“Large benign colon polyps treated with surgery”

 

Specific minor comment #13

Table 4: what does enolism stand for?

Response: We have now changed the term “active enolism” to “excessive alcohol consumption”, and define it as follows: “(≥30g/day in women and ≥40g/day in men)”.

Specific minor comment #14

Please put “per cent” together.

Response: Modified 

Discussion:

Specific minor comment #15

Please rephrase: “Computed tomography is the most profitable complementary test”; profitable does not seem appropriate for this instance.

Response: In response to both reviewers, we now speak of “yield” with regard to CT.

Specific minor comment #16

Figure 3: what do the figures below the graph represent (no. of patients at risk)? Is it the number of patients followed-up at the specific time intervals? It is not clear.

Response: The number of patients at risk is the number of patients alive at the start of each specific time interval. In the survival analysis, the difference in the number of patients between two consecutive time intervals corresponds to the patients who died or were lost to follow-up. Since there were no losses to follow-up in our study, it corresponds to the patients who died in this interval.

---

## [Decision Letter · Decision Letter 1]

9 Jun 2021

PONE-D-20-05824R1

Etiologies and 12-month mortality in patients with isolated involuntary weight loss at a rapid diagnostic unit

PLOS ONE

Dear Dr. Aligué,

Thank you for submitting your manuscript to PLOS ONE. After careful consideration, we feel that it has merit but does not fully meet PLOS ONE’s publication criteria as it currently stands. Therefore, we invite you to submit a revised version of the manuscript that addresses the points raised during the review process.

Reviewer 2 has raised some minor issues that need to be addressed before your manuscript can be accepted for publication.

We look forward to receiving your revised manuscript.

Kind regards,

Jamie Males

Staff Editor

PLOS ONE

Journal Requirements:

Reviewers' comments:

Reviewer's Responses to Questions

**Comments to the Author**

1. If the authors have adequately addressed your comments raised in a previous round of review and you feel that this manuscript is now acceptable for publication, you may indicate that here to bypass the “Comments to the Author” section, enter your conflict of interest statement in the “Confidential to Editor” section, and submit your "Accept" recommendation.

Reviewer #2: All comments have been addressed

Reviewer #3: (No Response)

2. Is the manuscript technically sound, and do the data support the conclusions?

Reviewer #2: Yes

Reviewer #3: Yes

3. Has the statistical analysis been performed appropriately and rigorously? 

Reviewer #2: Yes

Reviewer #3: Yes

4. Have the authors made all data underlying the findings in their manuscript fully available?

Reviewer #2: Yes

Reviewer #3: Yes

5. Is the manuscript presented in an intelligible fashion and written in standard English?

Reviewer #2: Yes

Reviewer #3: Yes

6. Review Comments to the Author

Reviewer #2: There are three minor comments:

- in the material and methods section, line 111, please change "it was not able" to "it was not possible".

- please review again the SD (standard deviations) for the amount of weight loss (kg) in Table 1 (in the first column SD in 4.7, while in all the others it is 0.3 and 0.8, which is somehow odd)

- in the discussion section, lines 58-59, please amend this formulation, since gastric, colonic and pancreatic are digestive cancers.

Reviewer #3: In this study, the authors use retrospective data collected at a single rapid diagnostic unit from years 2005-2013 of 791 patients presenting with involuntary weight loss above 5%. The cross-sectional visits are linked to diagnostic and mortality outcomes, allowing the authors to draw conclusions about IWL, disease, and mortality. The authors find, for example, that almost a quarter of patients with IWL were later diagnosed with cancer.

Thank you for the opportunity to read this interesting manuscript. I joined as a reviewer following the first set of revisions, and can see that both the reviewers and authors have already substantially improved the paper. I have just a few comments.

1. Percentage weight lost. In the list of variables collected, is initial/presenting weight measured/asked? How is % lost calculated? If the variable is available, it should be presented in the descriptive tables. This is especially true in illnesses that occur more in men/women. For example, since women weigh less than men on average, and are more likely to present with psychiatric illness, the % weight lost would be higher in this category. This would be interesting to see.

2. Brief question about “family support”. The phrase is referred to a few times in the manuscript. A brief phrase expanding what is meant by it (and why support is important in predicting RDU usage) would be helpful to the reader.

3. Previous research. While I wouldn’t usually ask an author to refer to my own work when writing a review, I think it could benefit the current manuscript. My co-authors and I are demographers and sociologists, so while the paper uses different survey and analytic methods, the findings could be useful for comparison with this manuscript and encourage cross-discipline dialogue. Though the papers differ--our paper, for example, used a nationally-representative survey that considered people who were diagnosed before retrospective survey, we didn’t exclude based on symptoms, and respondents didn’t need to present to a physician to confirm weight change—we also find evidence of the important role cancer plays in IWL.

Vierboom YC, Preston SH, Stokes A (2018) Patterns of weight change associated with

disease diagnosis in a national sample. PLoS ONE 13(11): e0207795. https://doi.org/10.1371/journal.pone.0207795

4. Will there be statistical programming code available to readers, alongside the data? It’s becoming customary in my field to make not just the data but also the code that produced the analysis and tables publicly available, to encourage transparency and reproducibility (of course, may be different in the medical field).

7. PLOS authors have the option to publish the peer review history of their article (what does this mean?). If published, this will include your full peer review and any attached files.

Reviewer #2: No

Reviewer #3: **Yes: **Yana Vierboom

---

## [Author Response · Author response to Decision Letter 1]

28 Jun 2021

Point by point response to the reviewers’ comments

Ref: PONE-D-20-05824R1 

“Etiologies and 12-month mortality in patients with isolated involuntary weight loss at a rapid diagnostic unit”

Reviewer #2: There are three minor comments:

MINOR CHANGES:

Specific minor comment #1

- in the material and methods section, line 111, please change "it was not able" to "it was not possible".

Response: Modified

Specific minor comment #2

- please review again the SD (standard deviations) for the amount of weight loss (kg) in Table 1 (in the first column SD in 4.7, while in all the others it is 0.3 and 0.8, which is somehow odd)

Response: We have detected a transcription error in the standard deviations of the variables age, weight loss (kg) previous 12 months, daily alcohol consumption (g/day) and Charlson Comorbidity Index. We have amended the values in the table 1. We apologize for the error.

Specific minor comment #3

- in the discussion section, lines 58-59, please amend this formulation, since gastric, colonic and pancreatic are digestive cancers.

Response: Modified. The percentage of digestive cancers is calculated over the total of diagnosed cancers. We have recalculated the percentage of gastric, colonic and pancreatic cancers over the total number of digestive cancers. In the previous version of the manuscript, the percentage was calculated based on the total number of cancers, which may have caused some confusion. 

 

Reviewer #3: In this study, the authors use retrospective data collected at a single rapid diagnostic unit from years 2005-2013 of 791 patients presenting with involuntary weight loss above 5%. The cross-sectional visits are linked to diagnostic and mortality outcomes, allowing the authors to draw conclusions about IWL, disease, and mortality. The authors find, for example, that almost a quarter of patients with IWL were later diagnosed with cancer.

Thank you for the opportunity to read this interesting manuscript. I joined as a reviewer following the first set of revisions, and can see that both the reviewers and authors have already substantially improved the paper. I have just a few comments.

Specific major comment #1

1. Percentage weight lost. In the list of variables collected, is initial/presenting weight measured/asked? How is % lost calculated? If the variable is available, it should be presented in the descriptive tables. This is especially true in illnesses that occur more in men/women. For example, since women weigh less than men on average, and are more likely to present with psychiatric illness, the % weight lost would be higher in this category. This would be interesting to see.

Response: In the anamnesis, the physician asked the patient about the number of kilos lost in the last 12 months, and about the initial/presenting weight in order to be able to know if the loss was equal or superior to 5%. However, information regarding the initial/presenting weight was not always registered in the patient's clinical records. Weight loss percentage was not included as a study variable. 

We agree that it would have been very interesting to have this information, but given the retrospective nature of the study, this was not possible.

Specific major comment #2

2. Brief question about “family support”. The phrase is referred to a few times in the manuscript. A brief phrase expanding what is meant by it (and why support is important in predicting RDU usage) would be helpful to the reader.

Response: One of the requirements for entering the RDU is the availability to attend additional visits and undergo tests on an outpatient basis in order to avoid hospital admission. Therefore, patients must be able to travel independently or otherwise have a primary caregiver who can accompany them.

Specific major comment #3

3. Previous research. While I wouldn’t usually ask an author to refer to my own work when writing a review, I think it could benefit the current manuscript. My co-authors and I are demographers and sociologists, so while the paper uses different survey and analytic methods, the findings could be useful for comparison with this manuscript and encourage cross-discipline dialogue. Though the papers differ--our paper, for example, used a nationally-representative survey that considered people who were diagnosed before retrospective survey, we didn’t exclude based on symptoms, and respondents didn’t need to present to a physician to confirm weight change—we also find evidence of the important role cancer plays in IWL.

Vierboom YC, Preston SH, Stokes A (2018) Patterns of weight change associated with

disease diagnosis in a national sample. PLoS ONE 13(11): e0207795. https://doi.org/10.1371/journal.pone.0207795

Response: The reviewer is totally justified in referring to her work as previous evidence of weight loss in a representative sample of the general adult population, according to their diagnosis with one of the six most prevalent pathologies either 0-1 years previously or 2+ years previously. 

There are discrepancies in the literature about whether weight loss varies depending on the etiology. Although we did not observe differences in the average weight lost in the last 12 months between different etiological groups, the study corroborates previous reports stating that cancer could be associated with a higher probability of unintentional weight loss. 

We have added this reference.

Specific major comment #4

4. Will there be statistical programming code available to readers, alongside the data? It’s becoming customary in my field to make not just the data but also the code that produced the analysis and tables publicly available, to encourage transparency and reproducibility (of course, may be different in the medical field).

Response: At the time we designed the study, we did not consider sharing patient data or statistical programming codes. However, if a researcher asks us for this information, we are happy to share it.

---

## [Decision Letter · Decision Letter 2]

10 Sep 2021

Etiologies and 12-month mortality in patients with isolated involuntary weight loss at a rapid diagnostic unit

PONE-D-20-05824R2

Dear Dr. Aligué,

We’re pleased to inform you that your manuscript has been judged scientifically suitable for publication and will be formally accepted for publication once it meets all outstanding technical requirements.

Kind regards,

Jamie Males

Staff Editor

PLOS ONE

Additional Editor Comments (optional):

Reviewers' comments:

Reviewer's Responses to Questions

**Comments to the Author**

1. If the authors have adequately addressed your comments raised in a previous round of review and you feel that this manuscript is now acceptable for publication, you may indicate that here to bypass the “Comments to the Author” section, enter your conflict of interest statement in the “Confidential to Editor” section, and submit your "Accept" recommendation.

Reviewer #2: All comments have been addressed

Reviewer #3: All comments have been addressed

2. Is the manuscript technically sound, and do the data support the conclusions?

Reviewer #2: Yes

Reviewer #3: Yes

3. Has the statistical analysis been performed appropriately and rigorously? 

Reviewer #2: Yes

Reviewer #3: Yes

4. Have the authors made all data underlying the findings in their manuscript fully available?

Reviewer #2: Yes

Reviewer #3: Yes

5. Is the manuscript presented in an intelligible fashion and written in standard English?

Reviewer #2: Yes

Reviewer #3: Yes

6. Review Comments to the Author

Reviewer #2: The authors addressed the previous comments in a concise and proper way. I think the manuscript was greatly improved so far.

Reviewer #3: Thank you for addressing my comments. I have no further concerns. Thank you for your valuable contribution.

7. PLOS authors have the option to publish the peer review history of their article (what does this mean?). If published, this will include your full peer review and any attached files.

Reviewer #2: No

Reviewer #3: No

---

## [Editor Report · Acceptance letter]

16 Sep 2021

PONE-D-20-05824R2 

Etiologies and 12-month mortality in patients with isolated involuntary weight loss at a rapid diagnostic unit 

Dear Dr. Aligué:

I'm pleased to inform you that your manuscript has been deemed suitable for publication in PLOS ONE. Congratulations! Your manuscript is now with our production department. 

Kind regards, 

on behalf of

Dr Jamie Males 

Staff Editor

PLOS ONE